# Conservative Uncertainty Estimation By Fitting Prior Networks

**Kamil Ciosek[1], Vincent Fortuin[1,2], Ryota Tomioka[1], Katja Hofmann[1], Richard Turner[1,3]**

## Abstract

Obtaining high-quality uncertainty estimates is essential for many applications of deep neural networks. In this paper, we theoretically justify a scheme for estimating uncertainties, based on sampling from a prior distribution. Crucially, the uncertainty estimates are shown to be conservative in the sense that they never underestimate a posterior uncertainty obtained by a hypothetical Bayesian algorithm. We also show concentration, implying that the uncertainty estimates converge to zero as we get more data. Uncertainty estimates obtained from random priors can be adapted to any deep network architecture and trained using standard supervised learning pipelines. We provide experimental evaluation of random priors on calibration and out-of-distribution detection on typical computer vision tasks, demonstrating that they outperform deep ensembles in practice.

## 1 Introduction

Deep learning has achieved huge success in many applications. In particular, increasingly often, it is used as a component in decision-making systems. In order to have confidence in decisions made by such systems, it is necessary to obtain good uncertainty estimates, which quantify how certain the network is about a given output. In particular, if the cost of failure is large, for example where the automated system has the capability to accidentally hurt humans, the availability and quality of uncertainty estimates can determine whether the system is safe to deploy at all (Carvalho, 2016; Leibig et al., 2017; Michelmore et al., 2018). Moreover, when decisions are made sequentially, good uncertainty estimates are crucial for achieving good performance quickly (Bellemare et al., 2016; Houthooft et al., 2016; Ostrovski et al., 2017; Burda et al., 2018).

Because any non-Bayesian inference process is potentially sub-optimal (De Finetti, 1937), these uncertainty estimates should ideally be relatable to Bayesian inference with a useful prior. Deep ensembles (Lakshminarayanan et al., 2017), one of the most popular methods available for uncertainty estimation in deep networks today, struggle with this requirement. While deep ensembles can be related (Rubin, 1981) to Bayesian inference in settings where the individual models are trained on subsets of the data, this is not how they are used in practice. In order to improve data efficiency, all ensembles are typically trained using the same data (Lakshminarayanan et al., 2017), resulting in a method which does not have a theoretical justification. Moreover, deep ensembles can give overconfident uncertainty estimates in practice. On the other hand, Monte-Carlo dropout can be viewed (Gal & Ghahramani, 2016) as a certain form of Bayesian inference. However, doing so requires requires either a limit to be taken or a generalization of variational inference to a quasi-KL divergence (Hron et al., 2018). In practice, MC dropout can give arbitrarily overconfident estimates (Foong et al., 2019). More broadly, a category of approaches, known as Bayesian Neural Networks (Blundell et al., 2015; Welling & Teh, 2011; Neal, 1996), maintains a distribution over the weights of the neural network. These methods have a sound Bayesian justification, but training them is both difficult and carries an accuracy penalty, particularly for networks with convolutional architectures (Osawa et al., 2019). Moreover, tuning BNNs is hard and achieving a good approximation to the posterior is difficult (Brosse et al., 2018).

We use another way of obtaining uncertainties for deep networks, based on fitting random priors (Osband et al., 2018; 2019). Random priors are easy to train and were found to work very well in practice (Burda et al., 2018). To obtain the uncertainty estimates,

Affiliations: 1. Microsoft Research Cambridge; 2. ETH Zurich; 3. University of Cambridge. The second author was an intern at Microsoft when contributing to this work.

we first train a predictor network to fit a prior. Two examples of prior-predictor pairs are shown in the top two plots of Figure 1.Faced with a novel input point, we obtain an uncertainty (Figure 1, bottom plot) by measuring the error of the predictor network against this pattern. Intuitively, these errors will be small close to the training points, but large far from them. The patterns themselves are drawn from randomly initialized (and therefore untrained) neural networks. While this way of estimating uncertainties was known before (Osband et al., 2019), it did not have a theoretical justification beyond Bayesian linear regression, which is too limiting for modern applications.

**Contributions** We provide a sound theoretical framework for obtaining uncertainty estimates by fitting random priors, a method previously lacking a principled justification. Specifically, we justify estimates in the uncertainty of the output *of neural networks with any architecture*. In particular, we show in Lemma 1 and Proposition 1 that these uncertainty estimates are *conservative*, meaning they are never more certain than a Bayesian algorithm would be. Moreover, in Proposition 2 we show concentration, i.e. that the uncertainties become zero with infinite data. Empirically, we evaluate the calibration and out-of-distribution performance of our uncertainty estimates on typical computer vision tasks, showing a practical benefit over deep ensembles and MC dropout.

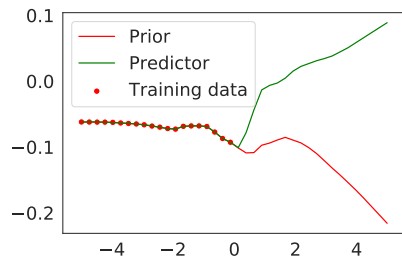

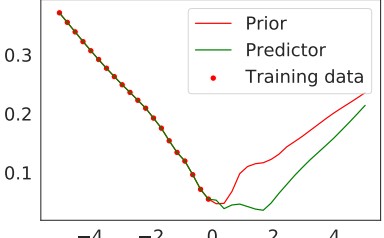

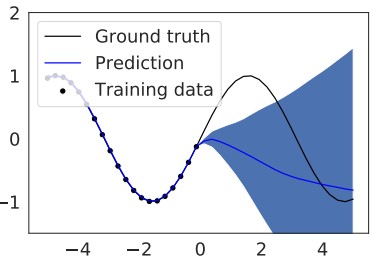

Figure 1: On top, two predictors (green) were trained to fit two randomly-generated priors (red). On the bottom, we obtain uncertainties from the difference between predictors and priors. Dots correspond to training points $x_i$.

## 2 PRELIMINARIES

We are going to reason about uncertainty within the formal framework of stochastic processes. We now introduce the required notations.

A *stochastic process* is a collection of random variables $\{f(x)\}$. We consider processes where $x \in \mathbb{R}^K$ and the random-variable $f(x)$ takes values in $\mathbb{R}^M$. A stochastic process has exchangeable outputs if the distribution does not change when permuting the $M$ entries in the output vector. Allowing a slight abuse of notation, we denote the finite-dimensional distribution of the process $\{f(x)\}$ for the set $X = \{x_i\}_{i=1,\ldots,N}$ as $f(x_1, \ldots, x_N) = f(X)$. In practice, the finite-dimensional distribution reflects the idea of restricting the process to points $x_1, \ldots, x_N$ and marginalizing over all the other points. Inference can be performed on stochastic processes similarly to probability distributions. In particular, we can start with some prior process $\{f(x)\}$, observe a set of $N$ training points $X = \{x_i\}_{i=1,\ldots,N}$ and labels $y = \{y_i\}_{i=1,\ldots,N}$ and then consider the posterior process $\{f_{Xy}(x)\}$, whose finite-dimensional distributions are given by $f_{Xy}(x_1^\star \ldots x_{N'}^\star) = f(x_1^\star \ldots x_{N'}^\star | x_1, \ldots, x_N, y_1, \ldots, y_N)$ for any set of testing points $x_1^\star \ldots x_{N'}^\star$. We use subscripts to denote conditioning on the dataset throughout the paper. We denote the variance of $f_{Xy}(x_\star)$ with $\sigma^2_{Xf}(x_\star)$. A stochastic process is called Gaussian if if all its finite-dimensional distributions are Gaussian. Given a test point $x_\star$, we denote the posterior GP mean with $\mu_{Xy}(x_\star)$ and posterior GP variance with $\sigma^2_X(x_\star)$. We provide more background on GPs in Appendix D.

## 3 ESTIMATING UNCERTAINTY FROM RANDOM PRIORS

**Intuition** Uncertainties obtained from random priors have an appealing intuitive justification. Consider the networks in the top part of Figure 1. We start with a randomly initialized prior network, shown in red. Whenever we see a datapoint, we train the predictor network (green) to match this

prior. Uncertainties can then be obtained by considering the squared error between the prior and the predictor at a given point. An example uncertainty estimate is shown as the shaded blue area in the bottom of Figure 1. While it may at first seem that the squared error is a poor measure of uncertainty because it can become very small by random chance, we formally show in Section 4.1 that this is very improbable. In Section 4.2, we show that this error goes down to zero as we observe more data. Similarly to GP inference, uncertainty estimation in our framework does not depend on the regression label. The prediction mean (blue curve in the bottom part of Figure 1) is obtained by fitting a completely separate neural network. In section 6, we discuss how this framework avoids the overconfidence characteristic of deep ensembles (Lakshminarayanan et al., 2017).

**Prior** The process of obtaining network uncertainties involves randomly initialized prior networks, *which are never trained*. While this may at first appear very different from they way deep learning is normally done, these random networks are a crucial component of our method. We show in Section 4.1 that the random process that corresponds to initializing these networks can be interpreted as a *prior* of a Bayesian inference procedure. A prior conveys the information about how the individual data points are related. The fact that we are using random networks has both practical and theoretical benefits. Practically, since the prior does not depend on the data, there is no way that it can overfit. The use of random priors also has strong empirical support – randomly initialized networks have been recently used as priors to obtain state-of-the-art performance on computer vision tasks (Ulyanov et al., 2018; Cheng et al., 2019). Theoretically, using random priors satisfies the *likelihood principle* (Robert, 2007). Moreover, random priors can be viewed as a safe choice since they make the minimum reasonable assumption that the network architecture is appropriate for the task. In fact, whenever deep learning is used, with or without uncertainty estimates, practitioners are already implicitly making that assumption.

**Algorithm** The process of training the predictor networks is shown in Algorithm 1. The function TRAIN-UNCERTAINTIES first generates random priors, i.e. neural networks with random weights. In our notation, it corresponds to sampling functions from the prior process $\{f(x)\}$. These priors, evaluated at points from the dataset $X = \{x_i\}_{i=1,\ldots,N}$ are then used as labels for supervised learning, performed by the function FIT. After training, when we want to obtain an uncertainty estimate $\phi$ at a given test point $x_\star$, we use the formula

$$\hat{\sigma}^2(x_\star) = \max(0, \hat{\sigma}_\mu^2(x_\star) + \beta\hat{v}_\sigma(x_\star) - \sigma_A^2). \quad (1)$$

---

**Algorithm 1** Training the predictors.

**function** TRAIN-UNCERTAINTIES($X$)
   **for** $i = 1 \ldots B$ **do**
      $f^i \sim \{f(x)\}$            ▷ random prior
      $h_{Xf^i} \leftarrow \text{FIT}(X, f^i(X))$
   **end for**
   **return** $f_i, h_{Xf^i}$
**end function**

**function** FIT($X, f^i(X)$)
   $L(h) \triangleq \sum_{x \in X} \|f^i(x) - h(x)\|^2$
   $h_{Xf^i} \leftarrow \text{OPTIMIZE}(L)$ ▷ SGD or similar
   **return** $h_{Xf^i}$    ▷ return trained predictor
**end function**

---

Here, the quantity $\hat{\sigma}_\mu^2$ is the sample mean of the squared error. We will show in Section 4 that it is an unbiased estimator of a variable that models the uncertainty. On the other hand, $\hat{v}_\sigma$ is the sample-based estimate of the standard deviation of squared error across bootstraps, needed to quantify our uncertainty about *what the uncertainty is*. The hyper-parameter $\beta$ controls the degree to which this uncertainty is taken into account. Formally, the quantities are defined as

$$\hat{\sigma}_\mu^2(x_\star) \triangleq \sum_{i=1}^{B} \frac{1}{MB} \|f(x_\star) - h_{Xf_i}(x_\star)\|^2, \quad (2)$$

$$\hat{v}_\sigma(x_\star) \triangleq \sqrt{\sum_{i=1}^{B} \frac{1}{B}(\hat{\sigma}_\mu^2(x_\star) - \frac{1}{M}\|f(x_\star) - h_{Xf_i}(x_\star)\|^2)^2}. \quad (3)$$

In the above equations, $B$ is the number of prior functions and each prior and predictor network has $M$ outputs. Because the predictors are trained independently, uncertainty estimates obtained from each of the $B$ predictor-prior pairs are independent. We defer the discussion of details of network architecture to Section 5. Our experiments (Section 7) show that it is often sufficient to use $B = 1$ in practice.

## 4 THEORETICAL RESULTS

In Section 3, we introduced a process for obtaining uncertainties in deep learning. We now seek to provide a formal justification. We define the expected uncertainties as

$$\tilde{\sigma}_\mu^2(x_\star) \triangleq \mathrm{E}_f\left[\hat{\sigma}_\mu^2(x_\star)\right] = \mathrm{E}_f\left[\frac{1}{M}\|f(x_\star) - h_{Xf}(x_\star)\|^2\right]. \tag{4}$$

In other words, $\tilde{\sigma}_\mu^2$ is the expected version of the sample-based uncertainties $\hat{\sigma}_\mu^2(x_\star)$ introduced in equation 2. Since Bayesian inference is known to be optimal (De Finetti, 1937; Jaynes, 2003; Robert, 2007), the most appealing way of justifying uncertainty estimates $\tilde{\sigma}_\mu^2$ and $\hat{\sigma}_\mu^2$ is to relate them to a Bayesian posterior $\sigma_{Xf}^2(x_\star)$. We do this in two stages. First, in Section 4.1, we prove that the obtained uncertainties are larger than ones arrived at by Bayesian inference. This means that our uncertainties are *conservative*, ensuring that our algorithm is never more certain than it should be. Next, in Section 4.2, we show that uncertainties concentrate, i.e., they become small as we get more and more data. These two properties are sufficient to justify the use of our uncertainties in many applications.

### 4.1 UNCERTAINTIES FROM RANDOM PRIORS ARE CONSERVATIVE

From the point of view of safety, it is preferable to overestimate the ground truth uncertainty than to underestimate it. We now show that this property holds for uncertainties obtained from random priors. We first justify conservatism for the expected uncertainty $\tilde{\sigma}_\mu^2$ defined in equation 4 and then for the sampled uncertainty $\hat{\sigma}_\mu^2$ defined in equation 2.

**Amortized Conservatism** We first consider a weak form of this conservatism, which we call amortized. It guarantees that $\tilde{\sigma}_\mu^2$ is never smaller than the average posterior uncertainty across labels sampled from the prior. Formally, amortized conservatism holds if for any test point $x_\star$ we have

$$\tilde{\sigma}_\mu^2(x_\star) \geq \mathrm{E}_{f(X)}\left[\sigma_{Xf}^2(x_\star)\right]. \tag{5}$$

Here $\sigma_{Xf}^2$ corresponds to the second moment of the posterior process $\{f_{Xf}(x)\}$. We will introduce a stronger version of conservatism, which does not have an expectation on the right-hand side, later in this section (eq. 8). For now, we concentrate on amortized conservatism. In Lemma 1 (proof in appendix), we show that it holds under very general conditions.

**Lemma 1.** *For any function $h : \mathbb{R}^{N \times (K+1)} \to \mathbb{R}^M$, for any test point $x_\star \in \mathbb{R}^K$ and for any stochastic process $\{f(x)\}_{x \in \mathbb{R}^K}$ with all second moments finite and exchangeable outputs*

$$\tilde{\sigma}_\mu^2(x_\star) = \mathrm{E}_{f(X)}\left[\sigma_{Xf}^2(x_\star) + \frac{1}{M}\|\mu_{Xf}(x_\star) - h_{Xf}(x_\star)\|^2\right]. \tag{6}$$

**Relation to a GP** Lemma 1 holds for any prior process $\{f(x)\}$. However, the prior process used by Algorithm 1 is not completely arbitrary. The fact that prior samples are obtained by initializing neural networks with independently sampled weights gives us additional structure. In fact, it can be shown that randomly initialized neural networks become close to GPs as the width of the layers increases. While the original result due to Neal (1996) held for a simple network with one hidden layer, it has been extended to a wide class of popular architectures, including to CNNs and RNNs of arbitrary depth (Matthews et al., 2018; Lee et al., 2018; Novak et al., 2019; Williams, 1997; Le Roux & Bengio, 2007; Hazan & Jaakkola, 2015; Daniely et al., 2016; Garriga-Alonso et al., 2019). Recently, it has been shown to hold for a broad class of functions trainable by gradient descent (Yang, 2019). While the precise statement of these results involves technicalities which fall beyond the scope of this paper, we recall the key insight. For a family of neural networks $\{f^W(x)\}$, where the weights are sampled independently and $W$ is the width of the hidden layers, there exists a limiting kernel function $k_\infty$ such that

$$\lim_{W \to \infty} [\{f^W(x)\}] = \mathcal{GP}(0, k_\infty). \tag{7}$$

In other words, as the size of the hidden layers increases, the stochastic process obtained by initializing networks randomly converges in distribution to a GP. In the context of our uncertainty estimates, this makes it reasonable for $W$ large enough to consider the prior to be a GP. We stress that the GP assumption has to hold *only for the prior network*, which is never trained. We do not make any assumptions about connections between the predictor training process and GPs.

**Strict Conservatism**    Denoting the posterior GP variance with $\sigma_X^2(x_\star)$, we define uncertainty estimates to be strictly conservative when

$$\tilde{\sigma}_\mu^2(x_\star) \geq \sigma_X^2(x_\star). \tag{8}$$

This statement is stronger than the amortized conservatism in equation 5. Intuitively, equation 8 can be interpreted as saying that our uncertainty estimates are never too small. This confirms the intuition expressed by Burda et al. (2018) that random priors do not overfit. Below, in Proposition 1, we outline how to guarantee strict conservatism formally. It is proved in Appendix F.1.

**Proposition 1** (Strict Conservatism in Expectation). *Assume that $f$ is a GP. Then for any function $h : \mathbb{R}^{N \times K} \to \mathbb{R}^M$, we have*

$$\tilde{\sigma}_\mu^2(x_\star) = \sigma_X^2(x_\star) + \underbrace{\mathrm{E}_{f(X)} \left[ \tfrac{1}{M} \| \mu_{Xf}(x_\star) - h_{Xf}(x_\star) \|^2 \right]}_{\geq 0}. \tag{9}$$

*Moreover, equality holds if and only if $h_{Xf}(x_\star) = \mu_{Xf}(x_\star)$.*

**Conservatism with Finite Bootstraps**    Lemma 1 above shows conservatism for expected uncertainties, i.e. $\tilde{\sigma}_\mu^2$ introduced in equation 5. However, in practice we have to estimate this expectation using a finite number of bootstraps, and use the sampled uncertainties $\hat{\sigma}_\mu^2$ defined in equation 2. We now state a conservatism guarantee that holds even in the case of just one bootstrap ($B = 1$). The proof is deferred to Appendix F.1.

**Corollary 1** (Strict Conservatism for Finite Bootstraps). *Assume that $f$ is a GP. Assume that the random variable $\hat{\sigma}_\mu^2(x_\star)$ has finite variance upper bounded by $v_{UB}$. Then with probability $1 - \delta$, for any function $h : \mathbb{R}^{N \times K} \to \mathbb{R}^M$, we have*

$$\hat{\sigma}_\mu^2(x_\star) + \tfrac{1}{\sqrt{\delta}} v_{UB} \geq \tilde{\sigma}_\mu^2(x_\star) \geq \sigma_X^2(x_\star). \tag{10}$$

However, applying Corollary 1 requires the knowledge of $v_{UB}$. We now provide an upper bound.

**Lemma 2.** *Assume that the GP $\{f(x)\}$ is zero mean with exchangeable outputs and the function $h_{Xf}$ takes values in $[-U, U]^M$. Assume that permuting the outputs of $f$ produces the same permutation in the outputs of $h_{Xf}$. With probability $1 - \delta$, we have*

$$\mathrm{Var}_{f_1, \ldots, f_B} \left[ \hat{\sigma}_\mu^2(x_\star) \right] \leq v_{UB}, \tag{11}$$

*where $v_{UB}$ is expressible in terms of observable quantities.*

The proof and the explicit formula for $v_{\mathrm{UB}}$ is deferred to Appendix F.1. In cases where conservatism is desired, but not absolutely essential, we can avoid the torturous calculation of Lemma 2 and replace $v_{\mathrm{UB}}$ with the sample-based estimate $\hat{v}_\sigma(x_\star)$, defined in equation 2. In this case, the conservatism guarantee is only approximate. This is how we obtained equation 1, used by the algorithm in practice.

## 4.2    Uncertainties from Random Priors Concentrate

While the conservatism property in Proposition 1 is appealing, it is not sufficient on its own for the uncertainty estimates to be useful. We also need concentration, i.e. a guarantee that the uncertainties $\hat{\sigma}^2$ become small with more data. We can gurantee this formally by assuming that the class of neural networks being fitted is Lipschitz-continuous and bounded. Intuitively, by assumption of Lipschitz continuity, the predictors $h_{Xf}$ cannot behave very differently on points from the training and test sets, since both come from the same data distribution. We can then show concentration by using standard Rademacher tools to obtain a bound on the expected uncertainty in terms of the squared error on the training set. This process is formalized in Proposition 2.

**Proposition 2.** *If the training converges, i.e. the training loss $\frac{1}{MN} \sum_{i=1}^{N} \| f(x_i) - h_{Xf}(x_i) \|^2 = \sigma_A^2$ for arbitrarily large training sets, then assuming the predictors $h_{Xf}$ are bounded and Lipschitz continuous with constant $L$, then under technical conditions the uncertainties concentrate, i.e. $\hat{\sigma}^2(x_\star) \to 0$ as $N \to \infty$ and $B \to \infty$ with probability 1.*

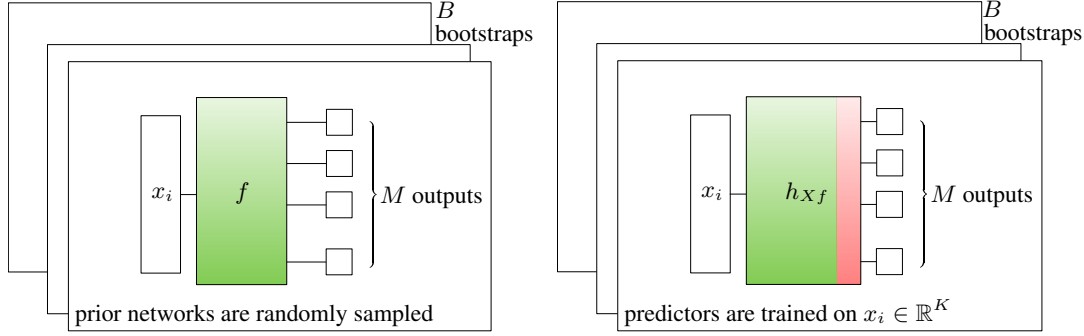

Figure 2: Architecture of the random prior networks $f$ and predictor networks $h_{Xf}$. The predictor networks $h_{Xf}$ typically share the same architectural core, but have additional layers relative to the prior networks. Both the green and red parts of the predictor networks are trained.

The proof and the technical conditions are given in Appendix F. Proposition 2 assumes that the training error is zero for arbitrarily large training sets, which might at first seem unrealistic. We argue that this assumption is in fact reasonable. The architecture of our predictor networks (Figure 2, right diagram) is a superset of the prior architecture (Figure 2, left diagram), guaranteeing the existence of weight settings for the predictor that make the training loss zero. Recent results on deep learning optimization (Du et al., 2019; Allen-Zhu et al., 2019) have shown that stochastic gradient descent can in general be expected to find representable functions.

## 5 PRACTICAL CONCLUSIONS FROM THE THEORY

We now re-visit the algorithm we defined in Section 3, with the aim of using the theory above to obtain practical improvements in the quality of the uncertainty estimates.

**Architecture and Choosing the Number of Bootstraps**  Our conservatism guarantee in Proposition 1 holds for *any* architecture for the predictor $h_{Xf}$. In theory, the predictor could be completely arbitrary and does not even have to be a deep network. In particular, there is no formal requirement for the predictor architecture to be the same as the prior. On the other hand, to show concentration in Proposition 2, we had to ensure that the prior networks are representable by the predictor. In practice, we use the architecture shown in Figure 2, where the predictor mirrors the prior, but has additional layers, giving it more representational power. Moreover, the architecture requires choosing the number of bootstraps $B$. Our experiments in Section 7 show that even using $B = 1$, i.e. one bootstrap, produces uncertainty estimates of high quality in practice.

**Modeling Epistemic and Aleatoric Uncertainty**  Proposition 1 and Proposition 2 hold for *any* Gaussian Process prior. By choosing the process appropriately, we can model both epistemic and aleatoric uncertainty. Denote by $\{n(x)\}$ a stochastic process obtained by randomly initializing neural networks and denote by $\{\epsilon(x)\sigma_A^2\}$ the noise term, modeling the aleatoric (observation) noise, where samples are obtained from $\epsilon(x) \sim \mathcal{N}(0,1)$ at each $x$ independently (see Appendix D for more background on aleatoric noise). We can now choose the prior process as a sum $\{f(x)\} = \{n(x) + \epsilon(x)\sigma_A^2\}$ of epistemic component $\{n(x)\}$ and the noise term. The amount of aleatoric uncertainty can be adjusted by choosing $\sigma_A^2$.

**Prior Choice, Weight Copying and Conservatism**  One question that can be asked about our architecture (Figure 2) is whether it is possible for the predictor to exactly copy the prior weights, giving zero uncertainty everywhere. A useful edge case to consider here is when we are solving a one-dimensional regression problem, $\sigma_A^2 = 0$ and the both the priors and predictors are linear functions. In this case, after training on two points, the predictors will agree with the priors everywhere and uncertainty estimates will be zero. However, this is still consistent with our conservatism guarantee The reason for this is once we assume such a linear prior, we are comparing to a GP *with a linear kernel*. But a GP with that kernel will also have zero uncertainty after seeing two samples.

In practice, this means that we have to choose the architecture of the prior networks be expressive enough, which is no different from choosing a reasonable prior for Bayesian inference. Empirically, the tested network architecture did not show weight copying.

## 6  PRIOR WORK

**Randomized Prior Functions (RPFs)**    Our work was inspired by, and builds on, Randomised Prior Functions (Osband et al., 2019; 2018), but it is different in two important respects. First, the existing theoretical justification for RPFs only holds for Bayesian linear regression (Osband et al., 2018, equation 3) with non-zero noise[1] added to the priors. In contrast, our results are much more general and hold *for any deep network* with or without added aleatoric noise. Second, we are targeting a different setting. While RPFs were designed as a way of sampling functions from the posterior, we provide estimates of posterior uncertainty at a given test point. Our algorithm is based on the work by Burda et al. (2018), who applied RPFs to exploration in MDPs, obtaining state-of-the art results, but without justifying their uncertainty estimates formally. Our paper provides this missing justification, while also introducing a way of quantifying the error in estimating the uncertainty itself. Moreover, since Burda et al. (2018) focused on the application of RPFs to Reinforcement Learning, they only performed out-of-distribution evaluation on the relatively easy MNIST dataset (LeCun, 1998). In contrast, in Section 7 we evaluate the uncertainties on more complex vision tasks. The term *prior networks* has also been used (Malinin & Gales, 2018) to denote deep networks that output the parameters of a prior distribution, an approach fundamentally different from our work.

**Deep Ensembles**    The main alternative approach for obtaining uncertainties in deep learning are deep ensembles (Lakshminarayanan et al., 2017). Building on the bootstrap (Efron & Tibshirani, 1994), deep ensembles maintain several models and quantify epistemic uncertainty by measuring how their outputs vary. Crucially, deep ensembles use representations trained on regression labels, and tend to learn similar representations for different inputs with similar labels, which can lead to over-fitting the uncertainty estimates. A useful edge case to consider is if the each of the models in the ensemble is convex in the weights. In this case, models in a deep ensemble will all converge to the same weights and produce zero uncertainty. While deep learning models used in practice aren't normally convex, we show empirically in section 7 that deep ensembles can give overconfident uncertainty estimates in practical vision tasks, particularly on points that have the same label as points in the training set. Since our method avoids overconfidence, it can be understood as complementary to deep ensembles, to be used in situations where obtaining conservative estimates is more important than the representational benefit of using labels. In practice, deep ensembles also require using more bootstraps to achieve the same OOD performance. Moreover, they do not have theoretical support in the case when all the members of the ensemble are trained on the same data, which is how they are used in practice (Lakshminarayanan et al., 2017).

**Dropout**    In cases where it is not economical to train more than one network, uncertainties can be obtained with dropout (Srivastava et al., 2014; Gal & Ghahramani, 2016). Monte-Carlo dropout can be viewed (Gal & Ghahramani, 2016) as a form of approximate Bayesian inference. However, to do so requires a rather unnatural approximating family from the perspective of approximate inference. Also, one has then either to take a limit or generalize variational inference to a quasi-KL (Hron et al., 2018) divergence. In addition, dropout can be interpreted in terms of MAP inference (Nalisnick et al., 2019). Another alternative view of MC dropout is as an ensemble method in which the ensemble members have shared parameters (which means they are trained together) and where the ensembling is applied at test time too. This latter view is arguably as natural as the Bayesian interpretation. For this reason we discuss MC dropout separately from BNNs. Since dropout implicitly approximates non-Gaussian weight distribution with Gaussians, it exhibits spurious patterns in the obtained uncertainties, which can lead to arbitrarily overconfident estimates (Foong et al., 2019). In contrast, due to the conservatism property, random priors avoid such overconfidence.

**Bayesian Neural Networks (BNNs)**    Bayesian Neural Networks (Blundell et al., 2015; Kingma & Welling, 2014; Rezende et al., 2014; Welling & Teh, 2011; Brosse et al., 2018) explicitly model the

---

[1]The existing justification of RPFs (Osband et al., 2019, Section 5.3.1) involves a division by the noise variance.

distribution over weights of a neural network. While BNNs provide a link between deep learning and Bayesian inference, they are very slow to train. Even recent tuned implementations of BNNs (Osawa et al., 2019) are several times slower than supervised learning. This happens despite using a battery of technical optimizations, including distributed training and batch normalization. Moreover, modern convolutional BNNs still carry a significant accuracy penalty when deployed with realistic settings of prior variance.[2]

# 7 EXPERIMENTS

Encouraged by the huge empirical success of random priors in Reinforcement Learning (Burda et al., 2018), we wanted to provide an evaluation in a more typical supervised learning setting. We tested the uncertainties in two ways. First, we investigated *calibration*, i.e. whether we can expect a higher accuracy for more confident estimates. Next, we checked whether the uncertainties can be used for out-of-distribution detection. We compared to two competing approaches for uncertainty detection: deep ensembles (Lakshminarayanan et al., 2017) and spatial concrete dropout (Gal et al., 2017). The same ResNet architecture served as a basis for all methods. Details of the implementation are provided in Appendix A.

**Out-Of-Distribution Detection** We evaluated the uncertainty estimates on out-of-distribution detection. To quantify the results, we evaluated the area under the ROC curve (AUROC) for the task of deciding whether a given image comes from the same distribution or not. All methods were trained on four classes from the CIFAR-10 (Krizhevsky et al., 2009) dataset (training details are provided in Appendix A). We then tested the resulting networks on images from withheld classes and on the SVHN dataset (Netzer et al., 2011), which contains completely different images. Results are shown in Table 1. Considering the statistical errors (see Appendix B), random priors performed slightly better than deep ensembles with adversarial training for $B = 1$ and about the same for $B = 10$. For dropout, $B$ refers to the number of dropout samples. Dropout performed worse, but was cheaper to train.

|  | RP | DE | DE +AT | DR |
|---|---|---|---|---|
| B=1 | | | | |
| Train v. cat/deer | **0.99** | 0.83 | 0.96 | 0.81 |
| Train v. vehicles | **1.00** | 0.82 | 0.96 | 0.76 |
| Train v. excluded | **1.00** | 0.82 | 0.96 | 0.77 |
| Train v. SVHN | **0.95** | 0.88 | **0.96** | 0.86 |
| B=10 | | | | |
| Train v. cat/deer | **1.00** | 0.95 | **0.99** | 0.82 |
| Train v. vehicles | **1.00** | 0.92 | 0.98 | 0.78 |
| Train v. excluded | **1.00** | 0.93 | 0.98 | 0.79 |
| Train v. SVHN | 0.97 | 0.94 | **0.99** | 0.87 |

Table 1: Out-of-distribution AUROC for random priors (RP), deep ensembles (DE), deep ensembles with adversarial training (DE+AT) and spatial concrete dropout (DR). Estimated confidence intervals are provided in Appendix B.

In order to gain a more finely-grained insight into the quality of the uncertainties, we also show uncertainty histograms in Figure 3. The figure shows the distribution of uncertainty estimates for seen data (top row) vs. unseen data (bottom row) for bootstrap sizes $B = \{1, 5, 10\}$. The main conclusion is that uncertainties obtained from random priors are already well-separated with $B = 1$, while deep ensembles need more bootstraps to achieve the full separation between test and train examples. We provide additional experimental results, showing OOD accuracy and an evaluation on CIFAR 100 in Appendix B.

**Calibration** Good uncertainty estimates have the property that accuracy increases as we become more certain, a property known as *calibration*. We measured it by evaluating average accuracy on the subset of images with uncertainty smaller than a given value. We trained on four classes from the CIFAR-10 (Krizhevsky et al., 2009) dataset. We then tested the resulting networks on the whole dataset, which included both the seen and unseen classes. Results are shown in Figure 4. Ideally, in a calibrated method, these curves should be increasing, indicating that a method always becomes more accurate as it becomes more confident. In coarse terms, Figure 4 confirms that all methods except a degenerate deep ensemble with only one bootstrap are roughly monotonic. However, uncertainty estimates from random priors are more stable, showing monotonicity on a finer scale as well as on a large scale. Interestingly, calibration improved only slightly when increasing the number of bootstraps $B$.

---

[2]See appendix E of the paper by Osawa et al. (2019).

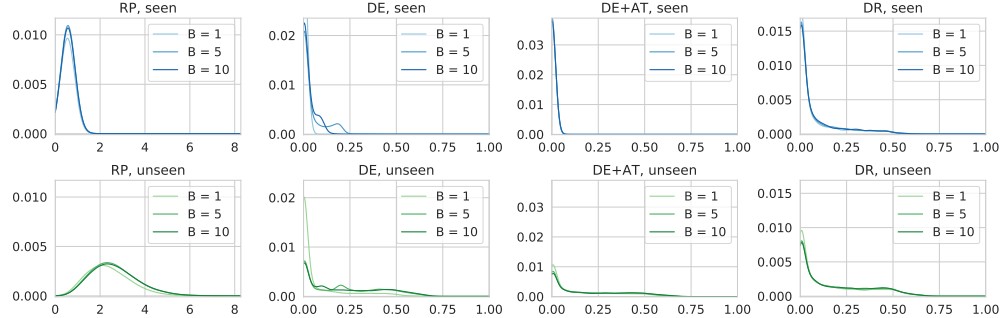

Figure 3: Distribution of uncertainty estimates for various algorithms. Top row shows seen data, bottom row shows unseen data from CIFAR-10. For random priors (RP), uncertainties are $\hat{\sigma}^2$. For other algorithms, they are $1 - \max(p_\mu)$, where $p_\mu$ is the averaged output of models in ensemble (Lakshminarayanan et al., 2017).

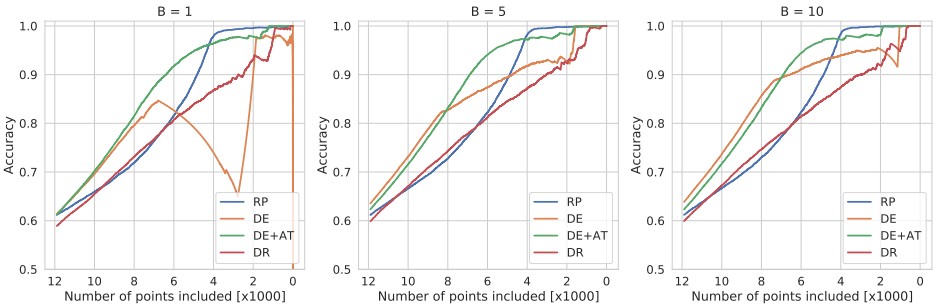

Figure 4: Calibration curves showing the relationship between uncertainty (horizontal axis) and accuracy (vertical axis) for $B = 1, 5, 10$ on CIFAR-10.

**Subsampling Ablation** In the previous experiment, we kept the architectural and optimization choices fixed across algorithms. This ensured a level playing field, but meant that we were not able to obtain zero training error on the predictor networks used by random priors. However, we also wanted to evaluate random priors in the setting of near-zero training error. To do this, we used a smaller set of training images, while still keeping the network architecture the same. This allowed us to obtain near-complete convergence (details in Appendix A).

|  | RP | DE | DE +AT | DR |
|---|---|---|---|---|
| B=1 | | | | |
| Train v. excluded | **1.00** | 0.90 | 0.89 | 0.91 |
| Train v. SVHN | **1.00** | 0.95 | 0.94 | 0.97 |
| B=10 | | | | |
| Train v. excluded | **1.00** | 0.94 | 0.90 | 0.92 |
| Train v. SVHN | **1.00** | 0.97 | 0.95 | 0.97 |

Table 2: Out-of-distribution AUROC for the same models as above (see Tab. 1) on subsampled data. Numbers are accurate up to $\pm 0.01$.

Results of this ablation are shown in Figures 5 and 6, as well as Table 2, analogous to our results on the full dataset presented above. In this sub-sampled regime, the random prior method easily outperformed competing approaches, showing better calibration (Fig. 5). The histograms in Figure 6 also demonstrate good separation between seen and unseen data. In the out-of-distribution benchmarks reported in Table 2, the random prior method has comfortably outperformed the baselines. While this training regime is not practical for real-life tasks, it demonstrates the potential performance of random priors when trained to full convergence.

**Sensitivity to Initialization Scale** We performed an ablation to test the robustness of our algorithm to the scaling of the weight initialization in the prior. Results are shown in Figure 7, where we plot the relationship between initialization scale (taken from the set $\{0.01, 0.1, 1.0, 2.0, 5.0, 10.0\}$) and AUROC performance on the CIFAR-10 task. OOD performance is relatively robust with respect to the weight initialization within one order of magnitude.

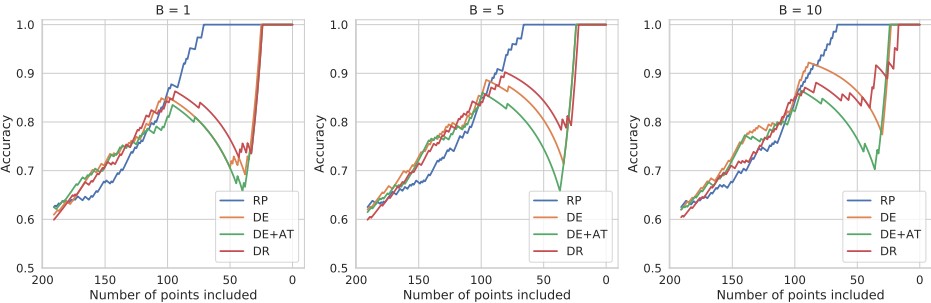

Figure 5: The relationship between uncertainty (horizontal axis) and accuracy (vertical axis) for $B = 1, 5, 10$ on a subset of 75 samples from CIFAR-10. In well-calibrated models, accuracy increases as uncertainty declines.

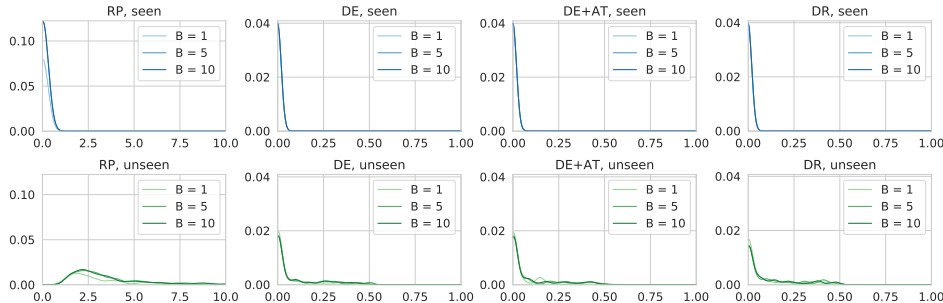

Figure 6: Distribution of uncertainty estimates for various algorithms. Top row shows seen data, bottom row shows unseen data from CIFAR-10, where we trained on a sample of 75 images from the training set. For random priors (RP), uncertainties are $\hat{\sigma}^2$. For other algorithms, they are $1 - \max(p_\mu)$, where $p_\mu$ is the averaged output of models in ensemble (Lakshminarayanan et al., 2017).

**Summary of experiments** We have shown that uncertainties obtained from random priors achieve competitive performance with fewer bootstraps in a regime where the network architecture is typical for standard supervised learning workloads. Random priors showed superior performance in a regime where the predictors can be trained to near-zero loss.

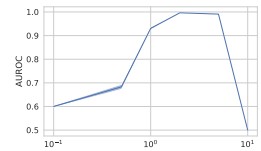

Figure 7: Robustness of OOD perfromance to initialization scale. Conf. bars present, but small, denoting high confidence. Horizontal axis is logarithmic.

## 8 CONCLUSIONS

We provided a theoretical justification for the use of random priors for obtaining uncertainty estimates in the context of deep learning. We have shown that the obtained uncertainties are conservative and that they concentrate for any neural network architecture. We performed an extensive empirical comparison, showing that random priors perform similarly to deep ensembles in a typical supervised training setting, while outperforming them in a regime where we are able to accomplish near-zero training loss for the predictors.

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

APPENDICES

APPENDIX A    REPRODUCIBILITY AND DETAILS OF EXPERIMENTAL SETUP

APPENDIX A.1    SYNTHETIC DATA

For the 1D regression experiment on synthetic data (Fig 1), we used feed-forward neural networks with 2 layers of 128 units each and a 1-dimensional output layer. We used an ensemble size of 5. The network was trained on 20 points sampled from the negative domain of a sigmoid function and tested on 20 points sampled from the positive domain.

APPENDIX A.2    EXPERIMENTAL SETUP

**Model architecture**    For the CIFAR-10 experiments, we adapted the setup from the `cifar10-fast` model.[3] For the network predicting the mean, we used the exact same architecture as in this model. For the prior networks in our uncertainty estimators, the architecture for the prior network was the same as the mean network, but using a final linear layer instead of the softmax layer. We used squared error on that last layer to get the uncertainties. For the predictor networks in the uncertainty estimators, we added two additional layers at the end to make sure the prior functions are learnable (see Fig. 2).

We followed Burda et al. (2018) in choosing the output size to be $M = 512$ and using the Adam optimizer (Kingma & Ba, 2014) with a learning rate of 0.0001. We optimized the initialization scale of our networks as a hyperparameter on the grid $\{0.01, 0.1, 1.0, 2.0, 10.0\}$ and chose 2.0. We chose a scaling factor of $\beta = 1.0$ for the uncertainty bonus of the random priors and fixed it for all experiments.

**Data**    For the CIFAR-10 experiment, we trained on the classes {bird, dog, frog, horse} and excluded {cat, deer, airplane, automobile, ship, truck}. For the small CIFAR-10 ablation experiment, we trained on 75 images sampled from the classes {ship, truck} and excluded the remaining classes.

**Training Error**    The training error was 0.57 ± 0.20 on the CIFAR experiment and 0.03 ± 0.02 on the sub-sampled ablation (the symbol ± denotes 90% confidence intervals).

**Out-of-distribution classification**    For computing the areas under the receiver-operator characteristic curves (AUROC) in the OOD classification tables, we used the `roc_auc_score` function from the Python package `sklearn` (Pedregosa et al., 2011), using the predicted uncertainties as predicted label scores and binary labels for whether or not the samples were from the training set.

APPENDIX B    ADDITIONAL RESULTS

APPENDIX B.1    CONFIDENCE INTERVALS FOR AUROCS

We provide confidence intervals for AUROC measurements in Table 3.

|  | RP | DE | DE +AT | DR |
|---|---|---|---|---|
| B=1 | | | | |
| Train v. cat/deer | **0.99 ± 0.002** | 0.83 ± 0.065 | 0.96 ± 0.008 | 0.81 ± 0.001 |
| Train v. vehicles | **1.00 ± 0.000** | 0.82 ± 0.070 | 0.96 ± 0.007 | 0.76 ± 0.001 |
| Train v. excluded | **1.00 ± 0.001** | 0.82 ± 0.069 | 0.96 ± 0.007 | 0.77 ± 0.002 |
| Train v. SVHN | **0.95 ± 0.013** | 0.88 ± 0.101 | **0.96 ± 0.009** | 0.86 ± 0.002 |

Table 3: Out-of-distribution AUROC for random priors (RP), deep ensembles (DE), deep ensembles with adversarial training (DE+AT) and spatial concrete dropout (DR). The errors are computed from ten samples each in the $B = 1$ case. The ± symbol denotes one standard error.

---

[3]`https://github.com/davidcpage/cifar10-fast`

## APPENDIX B.2  OOD CLASSIFICATION ACCURACIES

In addition to AUROC results, we also provide accuracy figures on the same OOD tasks. The thresholding for classification was obtained by cross-validation.

They are in Table 4 and 5.

| | RP | DE | DE +AT | DR |
|---|---|---|---|---|
| **B=1** | | | | |
| Train v. cat/deer | **0.97 ± 0.001** | 0.83 ± 0.008 | **0.97 ± 0.006** | 0.82 ± 0.000 |
| Train v. vehicles | **0.99 ± 0.001** | 0.81 ± 0.008 | 0.96 ± 0.004 | 0.86 ± 0.000 |
| Train v. excluded | **0.98 ± 0.001** | 0.87 ± 0.022 | 0.97 ± 0.007 | 0.70 ± 0.002 |
| Train v. SVHN | 0.91 ± 0.006 | 0.91 ± 0.025 | **0.96 ± 0.008** | 0.78 ± 0.001 |
| **B=10** | | | | |
| Train v. cat/deer | **0.98** | 0.88 | 0.96 | 0.82 |
| Train v. vehicles | **0.99** | 0.87 | 0.95 | 0.86 |
| Train v. excluded | **0.99** | 0.89 | 0.96 | 0.71 |
| Train v. SVHN | 0.92 | 0.88 | **0.96** | 0.78 |

Table 4: Out-of-distribution classification accuracy for random priors (RP), deep ensembles (DE), deep ensembles with adversarial training (DE+AT) and spatial concrete dropout (DR). These values augment the AUROC values reported in Table 1. The ± symbol denotes one standard error.

| | RP | DE | DE +AT | DR |
|---|---|---|---|---|
| **B=1** | | | | |
| Train v. excluded | **1.00** | 0.90 | 0.88 | 0.91 |
| Train v. SVHN | **1.00** | 0.95 | 0.90 | 0.97 |
| **B=10** | | | | |
| Train v. excluded | **1.00** | 0.95 | 0.89 | 0.91 |
| Train v. SVHN | **1.00** | 0.97 | 0.95 | 0.96 |

Table 5: Out-of-distribution accuracy for the same models as above (see Tab. 4) on subsampled data. These values augment the AUROC values reported in Table 2.

## APPENDIX B.3  SUPERVISED IN-DISTRIBUTION CLASSIFICATION ACCURACIES

| | RP* | DE | DE +AT | DR |
|---|---|---|---|---|
| CIFAR-10 | 0.86 | 0.88 | 0.86 | 0.86 |
| Subsampled CIFAR-10 | 0.82 | 0.81 | 0.82 | 0.75 |
| CIFAR-100 | 0.90 | 0.91 | 0.90 | 0.89 |

Table 6: In-distribution supervised classification accuracies on the respective test sets of the different data sets for random priors (RP), deep ensembles (DE), deep ensembles with adversarial training (DE+AT) and spatial concrete dropout (DR).
*Since random priors do not have an intrinsic supervised prediction model, we used the predictions from the DE+AT model in all our experiments instead, setting $B = 1$.

## APPENDIX B.4  CIFAR100 EXPERIMENT

As additional empirical support for our method, we ran experiments on another data set, namely CIFAR-100 (Krizhevsky et al., 2009). Again, we include 5 classes in the training set and exclude the remaining classes. The results are reported in the following (Figs. 8, 9; Tabs. 7, 8). They qualitatively and quantitatively support the same conclusions as our previous experiments.

## APPENDIX C  BACKGROUND ON BAYES RISK

For completeness, we recall the definition of Bayes Risk. We are often interested in minimizing the Mean Squared Error $E_f \left[ (f(x_\star) - w)^2 \right]$, where $x_\star$ is a given test point and $w$ is a variable we are

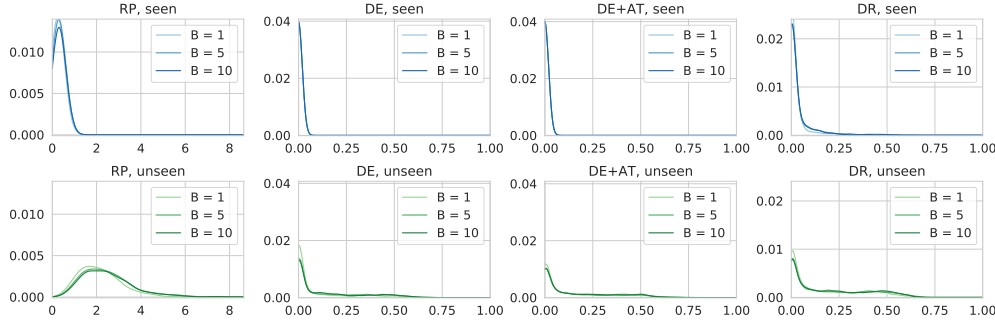

Figure 8: Distribution of uncertainty estimates for various algorithms. Top row shows seen data, bottom row shows unseen data from CIFAR-100. For random priors (RP), uncertainties are $\hat{\sigma}^2$. For other algorithms, they are $1 - \max(p_\mu)$, where $p_\mu$ is the averaged output of models in ensemble (Lakshminarayanan et al., 2017).

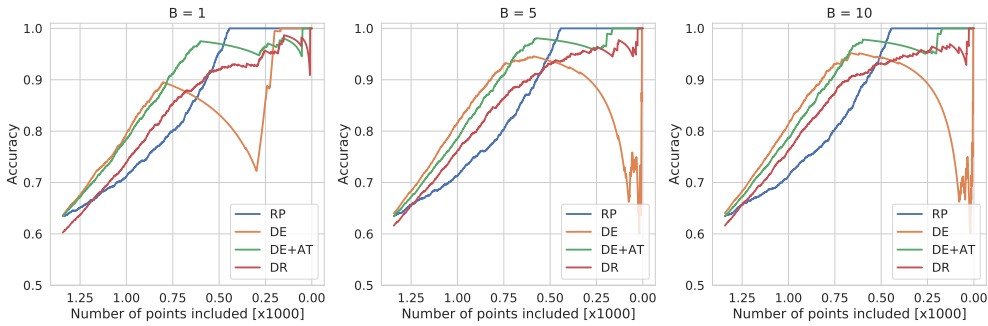

Figure 9: The relationship between uncertainty (horizontal axis) and accuracy (vertical axis) for $B = 1, 5, 10$ on samples from CIFAR-100. In well-calibrated models, accuracy increases as uncertainty declines.

allowed to adjust. A known result of Bayesian decision theory (Robert, 2007; Murphy, 2012) is that the minimizer of the MSE is given by the expected value of $\hat{f}$, i.e.

$$\arg\min_w \mathrm{E}_f \left[ (f(x_\star) - w)^2 \right] = \mathrm{E}_f [f(x_\star)]. \tag{12}$$

Equation 12 holds for any stochastic process $f$, including when $f$ is a posterior process obtained by conditioning on some dataset. A consequence of equation 12 is that it is impossible to obtain a MSE lower than the one obtained by computing the posterior mean of $f$.

## APPENDIX D  GAUSSIAN PROCESSES

A stochastic process is Gaussian (Williams & Rasmussen, 2006), if all its finite-dimensional distributions are Gaussian. The main advantage of GPs is that the posterior process can be expressed in a tractable way. GPs are often used for regression, where we are learning an unknown function[4] $\phi : \mathbb{R}^K \to \mathbb{R}$ from noisy observations. Since a Gaussian distribution is completely identified by its first two moments, a GP can be defined by a mean function and a covariance function. Formally, the notation $\mathcal{GP}(\mu, k)$ refers to a GP with with mean function $\mu : \mathbb{R}^K \to \mathbb{R}$, a positive-definite kernel function $k : \mathbb{R}^K \times \mathbb{R}^K \to \mathbb{R}$. GPs can be used to model two kinds of uncertainty: epistemic uncertainty, which reflects lack of knowledge about unobserved values of $\phi$ and aleatoric uncertainty, which reflects measurement noise. When performing regression, we start with a zero-mean prior $\mathcal{GP}(0, k)$ and then observe $N$ training points $X = \{x_i\}_{i=1,...,N}$ and labels $y = \{y_i\}_{i=1,...,N}$ where

---

[4]We depart from standard notation, which uses $f$, because we will be using $f$ to denote a sample from the prior process.

| | RP | DE | DE +AT | DR |
|---|---|---|---|---|
| B=1 | | | | |
| Train v. excluded | **1.00 ± 0.000** | 0.93 ± 0.003 | 0.98 ± 0.001 | 0.88 ± 0.002 |
| Train v. SVHN | **1.00 ± 0.000** | 0.96 ± 0.004 | 0.99 ± 0.001 | 0.82 ± 0.002 |
| B=10 | | | | |
| Train v. excluded | **1.00** | 0.96 | 0.99 | 0.90 |
| Train v. SVHN | **1.00** | 0.99 | **1.00** | 0.82 |

Table 7: Out-of-distribution classification AUROCs on CIFAR-100 for random priors (RP), deep ensembles (DE), deep ensembles with adversarial training (DE+AT) and spatial concrete dropout (DR). The ± symbol denotes one standard error.

| | RP | DE | DE +AT | DR |
|---|---|---|---|---|
| B=1 | | | | |
| Train v. excluded | **1.00 ± 0.001** | 0.91 ± 0.002 | 0.97 ± 0.001 | 0.82 ± 0.003 |
| Train v. SVHN | 0.97 ± 0.003 | 0.95 ± 0.003 | **0.99 ± 0.001** | 0.74 ± 0.003 |
| B=10 | | | | |
| Train v. excluded | **1.00** | 0.94 | 0.98 | 0.83 |
| Train v. SVHN | 0.98 | 0.98 | **0.99** | 0.74 |

Table 8: Out-of-distribution classification accuracy on CIFAR-100 for random priors (RP), deep ensembles (DE), deep ensembles with adversarial training (DE+AT) and spatial concrete dropout (DR). The ± symbol denotes one standard error. These values augment the AUROC values reported in Table 7.

$y_i = \phi(x_i) + \epsilon_i$. Here, the i.i.d. random variables $\epsilon_i \sim \mathcal{N}(0, \sigma_A^2)$ model the aleatoric noise. We obtain the posterior process on $\mathcal{GP}(\mu_{Xy}, k_X)$. For GPs, the mean and covariance of the posterior GP on $y$ evaluated at $x_\star$ can be expressed as

$$\mu_{Xy}(x_\star) = k_\star^\top (K + \sigma_A^2 I)^{-1} y \quad \text{and} \tag{13}$$

$$\sigma_X^2(x_\star) \triangleq k_X(x_\star, x_\star) + \sigma_A^2 = k_{\star\star} - k_\star^\top (K + \sigma_A^2 I)^{-1} k_\star + \sigma_A^2. \tag{14}$$

In particular, the posterior covariance does not depend on $y$. In the formula above, we use the kernel matrix $K \in \mathbb{R}^N \times \mathbb{R}^N$ defined as $K_{ij} = k(x_i, x_j)$, where $x_i$ and $x_j$ are in the training set. We also use the notation $k_\star \in \mathbb{R}^N$ for the vector of train-test correlations $\{k_\star\}_i = k(x_i, x^\star)$, where $x_i$ is in the training set and $k(x^\star, x^\star)$ is similarly defined. The shorthand $\sigma_X^2(x_\star)$ introduced in equation 14 denotes the posterior variance at a single point.

## APPENDIX E    LIST OF SYMBOLS DENOTING VARIANCE

Below, we give a list of symbols used for variance of various random variables.

| | |
|---|---|
| $\sigma_{Xf}^2$ | posterior variance of stochastic process |
| $\sigma_X^2$ | posterior variance of Gaussian process |
| $\sigma_0^2$ | prior variance of stochastic process |
| $\hat{\sigma}_0^2$ | sample-based estimate of prior GP variance |
| $\hat{\sigma}^2$ | combined uncertainty estimate (see equation 1) |
| $\hat{\sigma}_\mu^2$ | sample-based mean part of uncertainty estimate (see equation 2) |
| $\tilde{\sigma}_\mu^2$ | $\mathrm{E}_f \left[ \hat{\sigma}_\mu^2 \right]$ |
| $\hat{v}_\sigma$ | sample-based variance part of uncertainty estimate (see equation 3) |
| $v_{\mathrm{UB}}$ | upper bound on variance of $\hat{\sigma}_\mu^2$ |
| $\sigma_A^2$ | aleatoric variance (observation noise) |

## APPENDIX F    PROOFS

We now give formal proofs for the results in the paper.

APPENDIX F.1 PROOFS RELATING TO CONSERVATISM

**Lemma 1.** *For any function $h : \mathbb{R}^{N \times (K+1)} \to \mathbb{R}^M$, for any test point $x_\star \in \mathbb{R}^K$ and for any stochastic process $\{f(x)\}_{x \in \mathbb{R}^K}$ with all second moments finite and exchangeable outputs*

$$\tilde{\sigma}_\mu^2(x_\star) = \mathrm{E}_{f(X)}\left[\sigma_{Xf}^2(x_\star) + \tfrac{1}{M}\|\mu_{Xf}(x_\star) - h_{Xf}(x_\star)\|^2\right]. \tag{6}$$

*Proof.* We prove the statement by re-writing the expression on the left.

$$\tilde{\sigma}_\mu^2(x_\star) = \tfrac{1}{M}\,\mathrm{E}_{f(X),f(x_\star)}\left[\|f(x_\star) - h_{Xf}(x_\star)\|^2\right] \tag{15}$$

$$= \tfrac{1}{M}\,\mathrm{E}_{f(X)}\left[\mathrm{E}_{f(x_\star|f(X))}\left[\|f(x_\star) - h_{Xf}(x_\star)\|^2\right]\right] \tag{16}$$

$$= \tfrac{1}{M}\,\mathrm{E}_{f(X)}\left[\mathrm{E}_{f(x_\star|f(X))}\left[\sum_{m=1}^M (f^m(x_\star) - h_{Xf}^m(x_\star))^2\right]\right] \tag{17}$$

$$= \tfrac{1}{M}\,\mathrm{E}_{f(X)}\left[\mathrm{E}_{f(x_\star|f(X))}\left[\sum_{m=1}^M (f^m(x_\star))^2 - 2f^m(x_\star)h_{Xf}^m(x_\star) + (h_{Xf}^m(x_\star))^2\right]\right] \tag{18}$$

$$= \tfrac{1}{M}\,\mathrm{E}_{f(X)}\left[\sum_{m=1}^M \sigma_{Xf^m}^2(x_\star) + (\mu_{Xf^m}(x_\star))^2 - 2\mu_{Xf^m}(x_\star)h_{Xf}^m(x_\star) + (h_{Xf}^m(x_\star))^2\right] \tag{19}$$

$$= \tfrac{1}{M}\,\mathrm{E}_{f(X)}\left[\sum_{m=1}^M \sigma_{Xf^m}^2(x_\star) + (\mu_{Xf^m}(x_\star) - h_{Xf}^m(x_\star))^2\right] \tag{20}$$

$$= \mathrm{E}_{f(X)}\left[\sigma_{Xf}^2(x_\star) + \tfrac{1}{M}\|\mu_{Xf}(x_\star) - h_{Xf}(x_\star)\|^2\right] \tag{21}$$

Here, the equality in (16) holds by definition of conditional probability. The equality in (19) holds by definition of posterior mean and the equality 21 follows by assumption that the process has exchangeable outputs. While this argument follows a similar pattern to a standard result about Bayesian Risk (see Appendix Appendix C), it is not identical because the function $h_{Xf}$ depends on $f$. □

**Proposition 1** (Strict Conservatism in Expectation). *Assume that $f$ is a GP. Then for any function $h : \mathbb{R}^{N \times K} \to \mathbb{R}^M$, we have*

$$\tilde{\sigma}_\mu^2(x_\star) = \sigma_X^2(x_\star) + \underbrace{\mathrm{E}_{f(X)}\left[\tfrac{1}{M}\|\mu_{Xf}(x_\star) - h_{Xf}(x_\star)\|^2\right]}_{\geq 0}. \tag{9}$$

*Moreover, equality holds if and only if $h_{Xf}(x_\star) = \mu_{Xf}(x_\star)$.*

*Proof.* We instantiate Lemma 1 by setting $f$ to be a GP. By equation 14, the posterior covariance of a GP does not depend on the target values, i.e. $\sigma_{Xf}^2(x_\star) = \sigma_X^2(x_\star)$. The first part of the result can be shown by pulling $\sigma_X^2(x_\star)$ out of the expectation. Moreover, since $\|\cdot\|$ is a norm and hence positive semi-definite, equality holds if and only if $h_{Xf}(x_\star) = \mu_{Xf}(x_\star)$. □

**Lemma 3.** *Assume that the random variable $\hat{\sigma}_\mu^2(x_\star)$ has finite variance upper bounded by $v_{UB}$. With probability $1 - \delta$, we have $\hat{\sigma}_\mu^2(x_\star) + \tfrac{1}{\sqrt{\delta}} v_{UB} \geq \tilde{\sigma}_\mu^2(x_\star)$.*

*Proof.* The proof is standard, but we state it in our notation for completeness. Applying Chebyshev's inequality to the random variable $\hat{\sigma}_\mu^2(x_\star)$, we have that $\mathrm{Prob}\left(|\tilde{\sigma}_\mu^2(x_\star) - \hat{\sigma}_\mu^2(x_\star)| \geq \tfrac{1}{\sqrt{\delta}} v_{UB}\right) \leq \delta$, implying the statement. □

**Corollary 1** (Strict Conservatism for Finite Bootstraps). *Assume that $f$ is a GP. Assume that the random variable $\hat{\sigma}_\mu^2(x_\star)$ has finite variance upper bounded by $v_{UB}$. Then with probability $1 - \delta$, for any function $h : \mathbb{R}^{N \times K} \to \mathbb{R}^M$, we have*

$$\hat{\sigma}_\mu^2(x_\star) + \tfrac{1}{\sqrt{\delta}} v_{UB} \geq \tilde{\sigma}_\mu^2(x_\star) \geq \sigma_X^2(x_\star). \tag{10}$$

*Proof.* Combine Lemma 3 and Proposition 1. □

**Lemma 2.** *Assume that the GP $\{f(x)\}$ is zero mean with exchangeable outputs and the function $h_{Xf}$ takes values in $[-U, U]^M$. Assume that permuting the outputs of $f$ produces the same permutation in the outputs of $h_{Xf}$. With probability $1 - \delta$, we have*

$$\text{Var}_{f_1,\dots,f_B} \left[ \hat{\sigma}_\mu^2(x_\star) \right] \leq v_{UB}, \tag{11}$$

*where $v_{UB}$ is expressible in terms of observable quantities.*

*Proof.* We seek to decompose the variance of $\hat{\sigma}_\mu^2(x_\star)$ into the part that comes from the prior and the part that comes from the fitted function $h_{Xf^m}$.

$$\text{Var}_{f_1,\dots,f_B} \left[ \hat{\sigma}_\mu^2(x_\star) \right] \tag{22}$$

$$= \text{Var}_{f_1,\dots,f_B} \left[ \sum_{i=1}^B \frac{1}{MB} \| f(x_\star) - h_{Xf_i}(x_\star) \|^2 \right] \tag{23}$$

$$= \frac{1}{B} \text{Var}_f \left[ \frac{1}{M} \| f(x_\star) - h_{Xf_i}(x_\star) \|^2 \right] \tag{24}$$

$$= \frac{1}{B} \frac{1}{M^2} \text{Var}_f \left[ (\sum_{m=1}^M (f^m(x_\star) - h_{Xf^m}(x_\star))^2 \right] \tag{25}$$

$$= \frac{1}{B} \frac{1}{M^2} \sum_{m=1}^M \sum_{l=1}^M \text{Cov}_f \left[ (f^m(x_\star) - h_{Xf^m}(x_\star))^2, (f^l(x_\star) - h_{Xf^l}(x_\star))^2 \right] \tag{26}$$

$$\leq \frac{1}{B} \frac{1}{M^2} M^2 \text{Var}_f \left[ (f^m(x_\star) - h_{Xf^m}(x_\star))^2 \right] \tag{27}$$

$$= \frac{1}{B} \text{Var}_f \left[ (f^m(x_\star) - h_{Xf^m}(x_\star))^2 \right] \tag{28}$$

$$\leq \frac{1}{B} \text{E}_f \left[ (f^m(x_\star) - h_{Xf^m}(x_\star))^4 \right] \tag{29}$$

$$= \frac{1}{B} (\text{E}_f \left[ (f^m(x_\star))^4 \right] - 4 \text{E}_f \left[ (f^m(x_\star))^3 h_{Xf^m}(x_\star) \right] + 6 \text{E}_f \left[ (f^m(x_\star))^2 (h_{Xf^m}(x_\star))^2 \right]$$
$$- 4 \text{E}_f \left[ f^m(x_\star)(h_{Xf^m}(x_\star))^3 \right] + \text{E}_f \left[ (h_{Xf^m}(x_\star))^4 \right]) \tag{30}$$

Here, line 27 holds by exchangeability of outputs and the Cauchy-Schwarz inequality.

Since $h_{Xf^m}(x_\star)$ is has support in $[-U, U]$, we have

$$\text{E}_f \left[ (h_{Xf^m}(x_\star))^2) \right] \leq U^2, \;\; \text{E}_f \left[ (h_{Xf^m}(x_\star))^4) \right] \leq U^4, \;\; \text{E}_f \left[ (h_{Xf^m}(x_\star))^6) \right] \leq U^6. \tag{31}$$

Moreover, since $f(x_\star)$ is Gaussian and zero mean, we can write out the moments explicitly.

$$\text{E}_f \left[ (f^m(x_\star))^4 \right] = 3(\text{E}_f \left[ (f^m(x_\star))^2) \right])^2$$
$$\text{E}_f \left[ (f^m(x_\star))^6) \right] = 15(\text{E}_f \left[ (f^m(x_\star))^2) \right])^3 \tag{32}$$

Since $f(x_\star)$ is Gaussian, we can use a sample-based estimate of the prior variance and obtain an probabilistic confidence interval. In particular, we know that $\text{E}_f \left[ (f^m(x_\star))^2) \right] \leq \hat{\sigma}_0^2(x_\star) \frac{B_0-1}{\chi_I^2(\delta)}$ with probability $1 - \delta$, where $\chi_I^2$ denotes the inverse CDF of the Chi-Squared distribution with $B_0 - 1$ degrees of freedom. We denote this upper bound with $w_{UB} = \hat{\sigma}_0^2(x_\star) \frac{B_0-1}{\chi_I^2(\delta)}$.

We proceed by bounding the individual terms in equation 30 separately.

$$\text{E}_f \left[ (f^m(x_\star))^4 \right] = 3(\text{E}_f \left[ (f^m(x_\star))^2 \right])^2$$

$$- \text{E}_f \left[ (f^m(x_\star))^3 h_{Xf^m}(x_\star) \right] \leq \sqrt{\text{E}_f \left[ (f^m(x_\star))^6 \right] \text{E}_f \left[ (h_{Xf^m}(x_\star))^2 \right]}$$

$$\text{E}_f \left[ (f^m(x_\star))^2 (h_{Xf^m}(x_\star))^2 \right] \leq \sqrt{\text{E}_f \left[ (f^m(x_\star))^4 \right] \text{E}_f \left[ (h_{Xf^m}(x_\star))^4 \right]}$$

$$- \text{E}_f \left[ f^m(x_\star)(h_{Xf^m}(x_\star))^3 \right] \leq \sqrt{\text{E}_f \left[ (f^m(x_\star))^2 \right] \text{E}_f \left[ (h_{Xf^m}(x_\star))^6 \right]}$$

Combining the above, equation 30 and the bounds on individual moments in equations 31 and 32, we obtain

$$\text{Var}_{f_1,\dots,f_B} \left[ \hat{\sigma}_\mu^2(x_\star) \right] \leq \underbrace{\frac{1}{B} \left( 3 w_{UB}^2 + 4\sqrt{15 w_{UB}^3 U^2} + 6\sqrt{3 w_{UB}^2 U^4} + 4\sqrt{w_{UB} U^6} + U^4 \right)}_{v_{UB}}. \tag{33}$$

Here, $w_{UB} = \hat{\sigma}_0^2(x_\star) \frac{B_0-1}{\chi_I^2(\delta)}$, $\hat{\sigma}_0^2(x_\star)$ is a sample-based estimate of the prior variance obtained with $B_0$ samples, where $\chi_I^2$ denotes the inverse CDF of the Chi-Squared distribution with $B_0 - 1$ degrees of freedom.

□

APPENDIX F.2    PROOFS RELATING TO CONCENTRATION

We now proceed to the proofs showing concentration. We begin by formally defining a class of predictor networks.

**Definition 1** (Class $\mathcal{H}_U$ of Lipschitz networks)**.** *Consider functions $h : \mathbb{R}^K \rightarrow \mathbb{R}^M$. Let $j, j' = 1, \ldots, M$, index the outputs of the function. We define $\mathcal{H}_U$ so that each $h \in \mathcal{H}_U$ has the following properties for each $j, j'$.* **(P1)** *$h_j$ is Lipschitz continuous with constant $L$, i.e. $\|h_j(x) - h_j(x')\|_2 \leq L\|x - x'\|_2$ for all $x, x'$ with $\|x\|_\infty \leq 1$ and $\|x'\|_\infty \leq 1$,* **(P2)** *outputs are exchangeable, i.e. $\{h_j : h \in \mathcal{H}_U\} = \{h_{j'} : h \in \mathcal{H}_U\}$,* **(P3)** *the class is symmetric around zero, i.e. $h_j \in \{h_j : h \in \mathcal{H}_U\}$ implies $-h_j \in \{h_j : h \in \mathcal{H}_U\}$.* **(P4)** *$h_j$ is bounded, i.e. $\max_{\|x\|_\infty \leq 1} |h_j(x)| \leq U$.*

While the conditions in Definition 1 look complicated, they are in fact easy to check for predictor networks that follow the architecture in Figure 2. In particular, Lipschitz continuity **(P1)** has to hold in practice because its absence would indicate extreme sensitivity to input perturbations. Output exchangeability **(P2)** holds since reordering the outputs does not change our architecture. Symmetry around zero **(P3)** holds by flipping the sign in the last network layer. Boundedness **(P4)** is easy to ensure by clipping outputs. In the following Lemma, we obtain a bound on the expected uncertainty.

**Lemma 4.** *Consider a target function $f : \mathbb{R}^K \rightarrow \mathbb{R}^M$, where $j = 1, \ldots, M$, with the domain restricted to $\|x\|_\infty \leq 1$. Introduce a constant $U$ such that $\max_{\|x\|_\infty \leq 1} |f_j(x)| \leq U$. Denote the data distribution with support on $\{x : \|x\|_\infty \leq 1\}$ as $\mathcal{D}$. Moreover, assume $K \geq 3$. For $h_{Xf} \in \mathcal{H}_U$, with probability $1 - \delta$ we have*

$$\mathrm{E}_{x_\star \sim \mathcal{D}}\left[\tfrac{1}{M}\|f(x_\star) - h_{Xf}(x_\star)\|^2\right] \leq \tfrac{1}{MN}\sum_{i=1}^{N}\|f(x_i) - h_{Xf}(x_i)\|^2 + LU\, O\left(\tfrac{1}{\sqrt[K]{N}}\sqrt{\tfrac{\log(1/\delta)}{N}}\right). \tag{34}$$

*Proof.* The proof uses standard Rademacher tools. To avoid confusion across several conventions, we explicitly define the Rademacher complexity of a function class $\mathcal{G}$ as:

$$\hat{\mathfrak{R}}_N(\mathcal{G}) \triangleq \mathrm{E}_{u_i}\left[\sup_{g \in \mathcal{G}} \tfrac{1}{N}\sum_{i=1}^{N} u_i g(x_i)\right] = \mathrm{E}_{u_i}\left[\sup_{g \in \mathcal{G}} \tfrac{1}{N}\left|\sum_{i=1}^{N} u_i^j g(x_i)\right|\right]. \tag{35}$$

Here, the random variables $u_i$ are sampled i.i.d. using a discrete distribution with $\mathrm{Prob}(u_i = -1) = \mathrm{Prob}(u_i = 1) = \frac{1}{2}$ and the the second equality follows by using property **(P3)**. We start by applying the generic Rademacher bound (Mohri et al., 2018) to the function class $\mathcal{M} = \{x_1, \ldots, x_N, t_1 \ldots, t_N \rightarrow \frac{1}{U^2}\frac{1}{M}\|t_i - h(x_i)\|^2, h \in \mathcal{H}_U\}$, which contains the possible errors of the predictor.

$$\mathrm{E}_{x_\star \sim \mathcal{D}}\left[\tfrac{1}{B^2}\tfrac{1}{M}\|f(x_\star) - h_{Xf}(x_\star)\|^2\right]$$
$$\leq \tfrac{1}{MN}\tfrac{1}{B^2}\sum_{i=1}^{N}\|f(x_i) - h_{Xf}(x_i)\|^2 + \hat{\mathfrak{R}}_N(\mathcal{M}) + O\left(\sqrt{\tfrac{\log(1/\delta)}{N}}\right). \tag{36}$$

We now introduce the function class $\mathcal{M}' = \{x_1, \ldots, x_N, t_1 \ldots, t_N \rightarrow \frac{1}{B^2}(t_i^j - h^j(x_i))^2, h \in \mathcal{H}_U\}$, which models the per-output squared error. Because of property **(P2)**, $\mathcal{M}'$ does not depend on the output index $j$. By pulling out the sum outside the supremum in equation 35, we get

$$\hat{\mathfrak{R}}_N(\mathcal{M}) \leq \hat{\mathfrak{R}}_N(\mathcal{M}'). \tag{37}$$

by Talagrand's Lemma (Mohri et al., 2018; Duchi, 2009), we also have

$$\hat{\mathfrak{R}}_N(\mathcal{M}') \leq 4\hat{\mathfrak{R}}_N(\mathcal{H}_1). \tag{38}$$

Here, $\mathcal{H}_1 = \{\frac{1}{U}h^j : h \in \mathcal{H}_U\}$. By property **(P1)**, functions in $\mathcal{H}_1$ are Lipschitz continuous with constant $L/U$. Instantiating a known bound for Lipschitz-continuous functions (Luxburg & Bousquet, 2004, Theorem 18 and Example 4), and using the assumption $K \geq 3$, we get $\hat{\mathfrak{R}}_N(\mathcal{H}_1) \leq \frac{L}{U}\, O\left(\frac{1}{\sqrt[K]{N}}\right)$. The Lemma follows by combining this with equation 37 and equation 38, plugging into equation 36 and re-scaling by $U^2$. $\qquad\square$

Lemma 4 allowed us to relate the error on the training set to the expected error on the test set. It also shows that the two will be closer for small values of the Lipschitz constant $L$. We now use this Lemma to show our main concentration result (Proposition 2).

**Proposition 2.** *If the training converges, i.e. the training loss $\frac{1}{MN} \sum_{i=1}^{N} \|f(x_i) - h_{Xf}(x_i)\|^2 = \sigma_A^2$ for arbitrarily large training sets, then assuming the predictors $h_{Xf}$ are bounded and Lipschitz continuous with constant $L$, then under technical conditions the uncertainties concentrate, i.e. $\hat{\sigma}^2(x_\star) \to 0$ as $N \to \infty$ and $B \to \infty$ with probability 1.*

*Proof.* We are assuming the technical conditions of Lemma 4. Instantiating Lemma 4, setting the training loss to $\sigma_A^2$ in the RHS of equation 34 and letting $N \to \infty$, we obtain the following with probability 1:

$$\lim_{N \to \infty} \mathrm{E}_{x_\star \sim \mathcal{D}}[\hat{\sigma}_\mu^2(x_\star)] = \sigma_A^2. \tag{39}$$

This implies:

$$\lim_{N \to \infty} \mathrm{E}_{x_\star \sim \mathcal{D}}[\max(0, \hat{\sigma}_\mu^2(x_\star) - \sigma_A^2)] = 0. \tag{40}$$

From the continuity of $f$ and $h_{Xf}$ we have that $\hat{\sigma}_\mu^2$ is continuous in $x_\star$. Together with the property that the expression under the expectation is non-negative, this gives that for every $x_\star$.

$$\lim_{N \to \infty} \max(0, \hat{\sigma}_\mu^2(x_\star) - \sigma_A^2) = 0. \tag{41}$$

Since the right-hand side does not depend on $B$, we also have

$$\lim_{B \to \infty} \lim_{N \to \infty} \max(0, \hat{\sigma}_\mu^2(x_\star) - \sigma_A^2) = 0. \tag{42}$$

From the definition of $\hat{v}_\sigma$, we have that

$$\lim_{B \to \infty} \lim_{N \to \infty} \hat{v}_\sigma = 0. \tag{43}$$

We show the Lemma by combining equation 42 and equation 43 with equation 1. $\qquad \square$

