# OpenReview forum: "Conservative Uncertainty Estimation By Fitting  Prior Networks"
_ICLR.cc/2020/Conference — Accept (Poster)_

### Official Review · AnonReviewer1 · 2019-10-22
**Official Blind Review #1**

**Rating:** 6

**Review:**

This work introduces a simple technique to obtain uncertainty estimates for deep neural networks. This is achieved by having a set of random networks (i.e. neural networks where their parameters are randomly initialized) and then computing an uncertainty value based on the difference in the predictions between those random networks and networks that are trained to mimic them on a finite collection of points. The authors further show that this method results into uncertainties that are conservative, meaning that they are higher than the uncertainty of a hypothetical posterior, and concentrate, i.e. they converge towards zero when we get more and more data. The authors further draw connections to ensemble methods and discuss how such a method can be effectively realized in practice. They then evaluate their approach on an out-of-distribution detection task, they measure the calibration of their uncertainty estimates and finally perform a small ablation study for their concentration result.

This work is interesting as it seems to provide a simple way to obtain reasonable uncertainty estimates. For this reason it can potentially serve as a strong baseline for this field. The theoretical considerations also help in providing some guarantees about such an approach. Having said that, in my opinion the writing could use some more work in order to make things more clear as some critical experimental details and baselines are missing and thus do not make the method as convincing. Furthermore, I also believe that some clarifications on the theoretical aspects of this work, will help in boosting its quality. More specifically:

- How exactly do you apply your method on the classification scenario? Do you select an arbitrary hidden layer of the classification model for the prior and predictor network architectures or the output logits / softmax probabilities?  In appendix A you mention the architecture but not precisely how it is employed. I believe this can be an important piece of information in order to decipher the importance of e.g. the output dimensionality on the uncertainty quality, as higher dimensional outputs might be harder to approximate thus could induce a larger squared error and hence uncertainty.
- What is the average training error of the predictor networks for the out-of-distribution task and subsampling ablation task, i.e. how far away from concentration were the priors?
- An effect that I found weird is the following: what happens for the out-of-distribution examples when the predictor networks can perfectly predict the prior network outputs? Wouldn’t that then imply that the uncertainty would be zero for any input (even an out-of-distribution one), as the prior network and predictor network always agree? One could imagine that for e.g. simple priors and with sufficiently dense sampling of the domain of the function this can happen in practice.
- For the conservatism you show that your uncertainty estimate is higher, on average, than the posterior variance when you sample points from the model itself. In a sense this guarantee translates to the actual data when the prior is “correct”. How do those conservatism guarantees translate to the case when there is model misspecification, i.e. when the prior is not correct? Perhaps a small toy example would be informative.
- For the predictor networks as described in figure 2; do you train both the green and red parts of the network or only the red parts and keep the green part fixed to the values you used for the prior f? (This helps in understanding how easy / difficult is the task of the predictor network).
- What is the accuracy on the actual in-distribution prediction task for the RP and baselines? What did “B” correspond to for the dropout networks? Was it the number of dropout samples you averaged over to get the final predictive?
- How sensitive are the results on the actual initialization strategy of the prior network? It would be good to see e.g. some form of performance / init variance curve in order to decipher the sensitivity.

Other comments
- It is worth pointing out that [1] showed that Monte-Carlo dropout performs approximate MAP inference, which seems more plausible than the approximate Bayesian inference perspective of [2].
- In the introduction you argue that Bayesian neural networks rely on procedures different from standard supervised learning and thus most ML pipelines are not optimized for them in practice. Could you elaborate a bit about this statement? Variationally trained BNNs with e.g. the reparametrization trick [3, 4] are straightforward since you can just use backpropagation to update their (variational) parameters.
- What is the x-axis for Figure 3 for the baselines? (I take it that for RP it is the \hat{sigma}^2(x)).
- I believe that a comparison against a simple variationally trained BNN would make the results more convincing.

Misc
- Second page, “Figure 1, top two plota” -> “Figure 1, top two plots”
- Third page, “[…] introduced in equation 2 denotes the posterior covariance [….]” -> “[…] introduced in equation 2 denotes the posterior variance […]“
- Fifth page, “[…] this makes it is reasonable for W large enough […]” -> “[…] this makes it reasonable for W large enough […]”
- Sixth page, “Corollary 1 and proposition 2”; where is corollary 1? Do you mean Proposition 1?
- Seventh page, “[…] inspired by, an builds on, […]” -> “inspired by, and builds on, […]”
- Ninth page “montonicity” -> “monotonicity”

Overall, I tend to accept this work, although, depending on the author rebuttal and other discussions, I am willing to change my rating accordingly.

[1] Eric Nalisnick, José Miguel Hernández-Lobato, Padhraic Smyth, Dropout as a Structured Shrinkage Prior, 2019
[2] Yarin Gal, Zoubin Ghahramani, Dropout as a Bayesian Approximation: Representing Model Uncertainty in Deep Learning, 2016
[3] Diederik P. Kingma, Max Welling, Auto-Encoding Variational Bayes, 2014
[4] Danilo Jimenez Rezende, Shakir Mohamed, Daan Wierstra, Stochastic Backpropagation and Approximate Inference in Deep Generative Models, 2014

**Experience Assessment:**

I have published in this field for several years.

**Review Assessment: Checking Correctness Of Derivations And Theory:**

I assessed the sensibility of the derivations and theory.

**Review Assessment: Checking Correctness Of Experiments:**

I assessed the sensibility of the experiments.

**Review Assessment: Thoroughness In Paper Reading:**

I read the paper thoroughly.

---

> ### Author Response · Authors · 2019-11-08
> **Thanks for the detailed review!**
>
> We answer your individual major points below.
>
> "This work is interesting as it seems to provide a simple way to obtain reasonable uncertainty estimates. For this reason it can potentially serve as a strong baseline for this field"
> Thanks a lot. We appreciate.
>
> "the writing could use some more work in order to make things more clear"
> We will provide a revised version of the paper which we hope is clearer.
>
> "How exactly do you apply your method on the classification scenario? Do you select an arbitrary hidden layer of the classification model for the prior and predictor network architectures or the output logits / softmax probabilities?"
> For classification, the architecture for the prior network was the same as the classification network, but using a final linear layer instead of the softmax layer. We used squared error on that last layer to get the uncertainties. We will clarify this in the revised paper.
>
> "What is the average training error of the predictor networks for the out-of-distribution task and subsampling ablation task, i.e. how far away from concentration were the priors?"
> The histogram of the training error is the same as the histogram of the uncertainties (Figures 3 and 6), first column. We will provide a summary of training / test errors in the revised version of the paper.
>
> "An effect that I found weird is the following: what happens for the out-of-distribution examples when the predictor networks can perfectly predict the prior network outputs? Wouldn’t that then imply that the uncertainty would be zero for any input (even an out-of-distribution one), as the prior network and predictor network always agree? One could imagine that for e.g. simple priors and with sufficiently dense sampling of the domain of the function this can happen in practice."
> Our experiments show that exact network cloning does not happen in practice. On the theoretical side, a useful edge case to consider is if the priors are just linear functions and we are solving a one-dimensional regression problem. In this case, (assuming no observation noise), evaluating the prior at two points is enough to fit the linear function and completely know the value of the prior everywhere. As you say, afterwards, the prior and predictor will always agree and uncertainty estimates will be zero. However, this is still consistent with our theory! The reason for this is once we assume such a linear prior, we are comparing to a GP with a linear kernel. But a GP *with that kernel* will also have zero uncertainty after seeing two samples, hence the conservatism guarantee still holds. In practice, this means that we have to be careful when choosing the architecture of the prior networks However, this difficulty is no different from performing Bayesian inference with a prior inappropriate for the problem. Empirically, reasonable network architectures do not show such cloning behavior.
>
> "For the conservatism you show that your uncertainty estimate is higher, on average, than the posterior variance when you sample points from the model itself. In a sense this guarantee translates to the actual data when the prior is “correct”. How do those conservatism guarantees translate to the case when there is model misspecification, i.e. when the prior is not correct? Perhaps a small toy example would be informative."
> Our method is affected by model misspecification in the same way as Gaussian Processes are. In particular, our conservatism results guarantee that the uncertainty obtained from random priors are greater or equal than uncertainties obtained from a GP which uses the same prior. Also, since in practice the priors are neural networks, chosen to have large capacity, we are unlikely to end up in a situation where the prior has no support on the ground truth function.
>
> "For the predictor networks as described in figure 2; do you train both the green and red parts of the network or only the red parts and keep the green part fixed to the values you used for the prior f? (This helps in understanding how easy / difficult is the task of the predictor network)."
> In figure 2, we train the entire predictor networks (both parts). We will clarify the caption.
>
> "What is the accuracy on the actual in-distribution prediction task for the RP and baselines?"
> In the paper, we reported the AUROC as it is a more robust. measure of classification quality. In the revised version, we will additionally provide accuracy numbers in the appendix.
>
> "What did 'B' correspond to for the dropout networks? Was it the number of dropout samples you averaged over to get the final predictive?"
> As you say, B was the number of samples averaged over at test time. We will make this clearer in the revised submission.
>
> "How sensitive are the results on the actual initialization strategy of the prior network? It would be good to see e.g. some form of performance / init variance curve in order to decipher the sensitivity."
> We will try to do an ablation on the initialization scale.

---

> > ### Comment · AnonReviewer1 · 2019-11-14
> > **Response to rebuttal**
> >
> > Thank you for addressing thoroughly my comments and I appreciate the revised version of the manuscript. The clarity has improved and the extra sections help in providing more insight about potential edge cases of this method. One thing I noticed is that the classification accuracy numbers for the in-distribution test set for both the RP and baselines are still missing (there is Figure 4 but that is kind of coarse and I am not sure if the images are from the test set).
> >
> > For now, I am in general in favor of accepting this work so I will maintain my current grade. Should the authors manage to also get some convincing results against a simple variational BNN I will consider increasing my score.

---

> > > ### Author Response · Authors · 2019-11-14
> > > **Thanks for the comment.**
> > >
> > > The accuracy numbers have now been added (Appendix B2). We are working on a BNN example.

---

> > > ### Author Response · Authors · 2019-11-15
> > > **Thanks again for the detailed review - we have uploaded the final version.**
> > >
> > > Accuracy figures for the classification task are now in Appendix B.3. The training error is included in Appendix A.2 and accuracy figures for the OOD task are in Appendix B.2. The paper also now includes an ablation on the initialization scale. We tried training a BNN using the state-of-the art VOGN BNN optimizer (https://github.com/team-approx-bayes/dl-with-bayes) but couldn't get it to work with our ResNet architecture, despite very low learning rates, long training, applying several settings for the prior precision and generally putting as much effort into the BNN as the other methods. We understand this may not be everyone's opinion, but this may be taken as confirmation that current BNNs are just very hard to get to work with architectures combining residual connections and convolutions, necessary for modern vision tasks. In the end, we did not include the BNN results in the paper because we felt the architecture needs to be kept the same across methods for the experiments to be meaningful. We feel that thanks to our theoretical contributions, our paper still represents a useful contribution to the community.

---

> ### Author Response · Authors · 2019-11-08
> **Response to "Other Comments" and "Misc remarks".**
>
> Thanks again for the feedback.
>
> "It is worth pointing out that [1] showed that Monte-Carlo dropout performs approximate MAP inference, which seems more plausible than the approximate Bayesian inference perspective of [2]."
> Thanks for pointing this out, we will mention the interpretation of Dropout in terms of MAP inference and include [1 - Nalisnick et al.] in the revised version of the paper. We want to point out an importance difference in the setting - while our work talks about uncertainty of the posterior, MAP inference considers the effect of the prior on weights estimates.
>
> "In the introduction you argue that Bayesian neural networks rely on procedures different from standard supervised learning and thus most ML pipelines are not optimized for them in practice. Could you elaborate a bit about this statement? Variationally trained BNNs with e.g. the reparametrization trick [3, 4] are straightforward since you can just use backpropagation to update their (variational) parameters."
> While Bayesian Neural Networks provide a link between deep learning and Bayesian inference, they are very slow to train. Even recent tuned implementations of BNNs (Osawa et al.) are several times slower than supervised learning (see Table 1 in Osawa et al. for benchmarks in a paper that argues for BNNs). This happens despite using a battery of technical optimizations, including distributed training and batch normalization. Moreover, modern convolutional BNNs still carry a significant accuracy penalty when deployed with realistic settings of prior variance (see appendix E in Osawa et al.). We will include this point in the revised paper.
>
> "What is the x-axis for Figure 3 for the baselines? (I take it that for RP it is the \hat{sigma}^2(x))."
> We use the method for obtaining uncertainties recommended by the "deep ensembles" paper. The x axis shows (1 - p_max), where p_max is the probability of the most probable class. The probability vectors from each bootstrap are averaged over before performing the max.  We will clarify the description of the figure.
>
> "I believe that a comparison against a simple variationally trained BNN would make the results more convincing."
> We are not sure we would be able to run this experiment on time. We will report back.
>
> Also, thanks for pointing out the typos - we will fix this in the revised version.
>
> "Sixth page, “Corollary 1 and proposition 2”; where is corollary 1? Do you mean Proposition 1?"
> Yes, we meant Proposition 1. We will fix this.
>
> "Overall, I tend to accept this work"
> Thanks a lot for the detailed feedback. If you have any other suggestions you believe may improve the paper, please share!
>
> References (for this post):
> Osawa, K., Swaroop, S., Jain, A., Eschenhagen, R., Turner, R. E., Yokota, R., & Khan, M. E. (2019). Practical Deep Learning with Bayesian Principles.
> Eric Nalisnick, José Miguel Hernández-Lobato, Padhraic Smyth, Dropout as a Structured Shrinkage Prior, 2019

---

### Official Review · AnonReviewer2 · 2019-10-22
**Official Blind Review #2**

**Rating:** 6

**Review:**

Overview:
This paper introduces a new method for uncertainty estimation which utilizes randomly initialized networks. Essentially, instead of training a single predictor that outputs means and uncertainty estimates together, authors propose to have two separate models: one that outputs means, and one that outputs uncertainties. The later one consists of two networks: a randomly initialized “prior” which is fixed and is not trained, and a “predictor”, which is then trained to predict the output of the randomly initialized “prior” applied to the training samples.
Authors show that under some reasonable assumptions the resulting estimates are conservative and concentrated (i.e. bounded and converge to zero with more data).

Writing quality:
Overall, the paper is relatively well-written, although it might be at times hard to follow, especially for someone who is not familiar with the original work that used randomized prior functions (Burda’18, Osband ‘18, ‘19).

Evaluation:
The method is experimentally evaluated on a task of out-of-distribution detection on CIFAR+SVHN, and seems to perform on-par or better than the baselines (including “standard” deep ensembles and dropout networks). In addition, there are experiments that demonstrate that the model is performing relatively well in terms of calibration (whether the model predictive behaviour makes sense as the model confidence changes).

Decision:
I find the core idea behind the paper quite interesting, however, as indicated by authors themselves, it has already been studied in a slightly different context (RL, works by Burda et. al, Osband et. al). That said, authors do provide additional insides for the supervised settings, and also analyse theoretically the behaviour of uncertainty estimates.
Overall, I cannot say I am fully convinced that the paper should be accepted as is (also see questions below), but generally I am positive about this work, and hence the final score: “weak accept”.

Additional comments / questions:
(somewhat minor) p1: “While deep ensembles …, where the individual ensembles are trained on different data“ - here and related text, it should probably be “individual models” / “individual networks”. Generally, I am not convinced that these are strong arguments against deep ensembles.

(minor) p2-p3: “2. Preliminaries” - I am not sure if this section adds much to the understanding, it would seem more natural to spend more time explaining the intuitions behind the net

(kind of major) p3. “prior” - The explanation of why using a randomly initialized network makes sense is not very strict. I kind of get the general idea, but it is not clear to me why not use something less expensive, e.g. just random projections, and why do we actually need a full network. Intuitively it seems quite strange to waste a lot of capacity to fit to essentially fit a set of random weights: is it something that allows the network to avoid easily learning the “random prior”? And, more generally, can this also be considered as a “trick” to de-correlate individual predictors? I believe these points should be discussed in more detail.

<update>
I would like to thank authors for verbose response and the revised version: it is a bit more clear.
I stand by my original rating.
</update>








**Experience Assessment:**

I have read many papers in this area.

**Review Assessment: Checking Correctness Of Derivations And Theory:**

I assessed the sensibility of the derivations and theory.

**Review Assessment: Checking Correctness Of Experiments:**

I assessed the sensibility of the experiments.

**Review Assessment: Thoroughness In Paper Reading:**

I read the paper thoroughly.

---

> ### Author Response · Authors · 2019-11-08
> **Thanks for the review.**
>
> We address your points below.
>
> "Overall, the paper is relatively well-written, although it might be at times hard to follow, especially for someone who is not familiar with the original work that used randomized prior functions (Burda’18, Osband ‘18, ‘19)."
> Thanks for the kind words! We will shortly upload a revised version of the paper with improved clarity.
>
> "I find the core idea behind the paper quite interesting, however, as indicated by authors themselves, it has already been studied in a slightly different context (RL, works by Burda et. al, Osband et. al). That said, authors do provide additional insides for the supervised settings, and also analyse theoretically the behaviour of uncertainty estimates."
> Thanks for pointing this out. The main contribution of the paper is a theoretical analysis of uncertainties obtained from fitting random priors for *arbitrary deep networks*. This is different from Osband's work, which discusses Bayesian linear regression. We believe that the difference in setting is hugely significant, because non-linear function approximation is necessary for many applications of Machine Learning. The work by Burda et al. provides great empirical results in RL, but no theoretical justification for the uncertainties.
>
> "(somewhat minor) p1: 'While deep ensembles …, where the individual ensembles are trained on different data' - here and related text, it should probably be 'individual models' / 'individual networks'. Generally, I am not convinced that these are strong arguments against deep ensembles."
> You are right about the wording - we meant 'individual network'. In terms of comparing our work to deep ensembles, we wanted to make two points. First, our work can be understood as complementary, to be used in situations where the conservatism property is desired. Second, our method has some practical advantages over deep ensembles. In practice, deep ensembles can also become overconfident (Figure 6) and poorly calibrated (Figure 5). Also, deep ensembles require more bootstraps (we will describe this more explicitly in the updated paper). We will provide a more detailed comparison with deep ensembles in the revised version of the paper. Also, the main theoretical argument against deep ensembles is that the uncertainties obtained from them in practice (when each ensemble was trained on the whole dataset) do not have a connection to a Bayesian posterior.
>
> "(minor) p2-p3: “2. Preliminaries” - I am not sure if this section adds much to the understanding, it would seem more natural to spend more time explaining the intuitions behind the net"
> In the revised submission, we will move some of the preliminary material to the appendix and use the space to provide more intuitions.
>
> "The explanation of why using a randomly initialized network makes sense is not very strict. I kind of get the general idea, but it is not clear to me why not use something less expensive, e.g. just random projections, and why do we actually need a full network. Intuitively it seems quite strange to waste a lot of capacity to fit to essentially fit a set of random weights: is it something that allows the network to avoid easily learning the “random prior”? And, more generally, can this also be considered as a “trick” to de-correlate individual predictors? I believe these points should be discussed in more detail."
> Thanks for pointing this out. On the formal side, the justification for using random priors is that our uncertainty estimates overestimate the Gaussian posterior. Deep ensembles do not have a comparable guarantee. Intuitively, the role that the random networks play is to avoid spurious confidence. We would not say that random priors waste capacity since the predictor network still depends on the x's (training points) and that is very useful for OOD detection. The fact that the predictors are trained independently means, as you say, that uncertainty estimates obtained from each predictor-prior pair are independent (de-correlated). We will include these insights in the revised version of the paper.
>
> Overall, thanks for the detailed comments. If you have other suggestions for improving the paper, please share!

---

> ### Author Response · Authors · 2019-11-15
> **Thanks again for the review. We summarise the relevant changes made in the final revision.**
>
> Relative the the original submission. We added an extended discussion of the prior and extended the comparison with deep ensembles, describing an explicit situation when they fail (each model in ensemble convex in weights). We shortened the preliminaries and addressed the clarity issues. Our key contribution over from prior work by Osband is that we deal with non-linear function approximation, a key requirement for many practical tasks (Osband's work only had theory for linear regression).

---

### Official Review · AnonReviewer3 · 2019-10-23
**Official Blind Review #3**

**Rating:** 6

**Review:**

The paper shows that the MSE of a deep network trained to match fixed random network is a conservative estimate of uncertainty in expectation over many such network pairs. An experiment compares this previously proposed method to other approaches for uncertainty estimation on CIFAR 10.

Strengths:
- Obtaining uncertainty estimates for predictions of deep neural networks is an important and open research question.
- Proposition 1 is an interesting result, although the paper does not seem to discuss its significance and implications enough.

Weaknesses:
- Proposition 1 is described as "our uncertainty estimates are never too small". However, as the name of the proposition suggests, it seems to only hold in expectation over models trained, which is a quite different statement.
- Proposition 2 seems to simply state that a small network can be distilled into a large network. Maybe I missed part of the reasoning here? Otherwise, it should probably not be highlighted as a major contribution.
- The experimental evaluation is very limited, training only on CIFAR 10. While the experiments add little value to the community, this may be acceptable for a mostly theoretical paper.

Clarity:
- The paper was difficult to read and unclear in explanations. It could help to define notation upfront instead of introducing shorthands on the way.
- In Figure 3, the Y axis limits should be fixed across seen and unseen histograms for the same method. The current presentation is a bit misleading here, as the presented method seems to have moved the most mass under this chart scaling.

Comments:
- As found in prior work cited in the submission, the method tends to perform well with just one network pair. This raises the question whether the contribution of the paper that holds in expectation over many pairs and the empirical success of the approach are connected.
- The marginal posterior variance \sigma^2(x_\star) appears in various forms with hat, tilde, and different subscripts. It maybe worth assigning different letters to these to avoid confusion.

**Experience Assessment:**

I have published one or two papers in this area.

**Review Assessment: Checking Correctness Of Derivations And Theory:**

I assessed the sensibility of the derivations and theory.

**Review Assessment: Checking Correctness Of Experiments:**

I assessed the sensibility of the experiments.

**Review Assessment: Thoroughness In Paper Reading:**

I read the paper at least twice and used my best judgement in assessing the paper.

---

> ### Author Response · Authors · 2019-11-08
> **Thanks for the review.**
>
> Below, we address your points in detail.
>
> "[in the Strength Section] Proposition 1 is an interesting result, although the paper does not seem to discuss its significance and implications enough".
> Overall, the main contribution of the paper is the we provide a sound theoretical framework for what was previously an ad-hoc method lacking principled justification. We will provide more background specifically about the usefulness Proposition 1 in a revised version of the paper. The main justification is that Proposition 1 guarantees that we won't become overconfident. We show empirically (Figures 3 and 6) that this problem can happen when using deep ensembles.
>
> "Proposition 1 is described as 'our uncertainty estimates are never too small'. However, as the name of the proposition suggests, it seems to only hold in expectation over models trained, which is a quite different statement."
> You are right when you say that Proposition 1 on its own holds in expectation. However, we also provide a result for a finite number of trained models in Lemmas 2 and 3 in the "Finite Bootstraps" section of the paper. Combining these two Lemmas with Proposition 1 gives a conservatism guarantee that is valid even for just one trained model. We agree that it is not very clearly presented - we will will add an explicit corollary that provides a conservatism guarantee with a finite number of bootstraps / models.
>
> "Proposition 2 seems to simply state that a small network can be distilled into a large network. Maybe I missed part of the reasoning here? Otherwise, it should probably not be highlighted as a major contribution."
> Concentration is a crucial property for uncertainty estimates and we feel that a paper without this result would be incomplete. Proposition 2 shows that, given enough data, uncertainties become small on *unseen* points. This is not the same as simply distilling the smaller "prior" network on some dataset. In other words, in order for the concentration result to hold, the capacity of the larger (predictor) network has to be controlled (which we do by the Lipschitz constant). While the Rademacher complexity tools we use to show this result are standard, to the best of our knowledge, they have not been used in a similar context.
>
> "The experimental evaluation is very limited, training only on CIFAR 10. While the experiments add little value to the community, this may be acceptable for a mostly theoretical paper."
> As you say, the experiments are meant to lend support to the theoretical part of the paper. The experimental evaluation has been carefully selected to focus on how conservative uncertainty estimates avoid overconfidence. For example, Figures 3 and 6 show that uncertainty estimates obtained from random priors show a better separation between seen and unseen points. They are also better calibrated - Figure 5 shows that the curves for random priors are closer to monotonic than for competing approaches. We believe that our experiments do provide a value for the community in the sense of validating theory.
>
> "The paper was difficult to read and unclear in explanations. It could help to define notation upfront instead of introducing shorthands on the way."
> Thanks for pointing this out. We will upload a revised submission where the notation is clearer.
>
> "In Figure 3, the Y axis limits should be fixed across seen and unseen histograms for the same method. The current presentation is a bit misleading here, as the presented method seems to have moved the most mass under this chart scaling"
> We will revise Figure 3 (and also Figure 6) to allow a direct comparison. We agree that it will make the figures easier to interpret.
>
> "[In the comments] As found in prior work cited in the submission, the method tends to perform well with just one network pair. This raises the question whether the contribution of the paper that holds in expectation over many pairs and the empirical success of the approach are connected."
> Thanks to Lemmas 2 and 3, out theory carries over to a finite number of prior-predictor pairs (including just one pair). We will add a corollary to make this explicit.
>
> "The marginal posterior variance \sigma^2(x_\star) appears in various forms with hat, tilde, and different subscripts. It maybe worth assigning different letters to these to avoid confusion."
> We will provide a revised version of the paper with an updated notation.
>
> Overall, thanks a lot for the feedback. Please let us know if you have other suggestions for improving the paper.

---

> ### Author Response · Authors · 2019-11-15
> **Thanks again for the review - we summarise the major relevant changes in the final revision.**
>
> Our conservatism guarantee holds for finite bootstraps as well as in expectation. We clarified this by explicitly adding Corollary 1. We believe Proposition 1 is significant because uncertainty estimates that avoid overconfidence .
> We fixed the clarity issues you mentioned. We enhanced the experimental evaluation by adding a new experiment on CIFAR100 (appendix B.4). We feel this is sufficient for the paper to meaningfully contribute to the community.

---

### Public Comment · ~Anthony_Wittmer1 · 2019-09-27
**A closely related paper**

Great work and I really enjoy reading it.

However, previous work has also studied the uncertainty estimation via prior networks. Please check out this paper [1]

In my opinion, a discussion/comparison seems due.


[1] Predictive Uncertainty Estimation via Prior Networks, NeurIPS 2018

---

> ### Author Response · Authors · 2019-09-30
> **Thanks for the pointer.**
>
> Wow, thanks for the pointer. You are right that a discussion of [1] is due. Both works use the term "prior networks", but they use it to mean different things - [1] learns a network that parameterizes a Dirichlet prior, while our work uses fixed prior networks to obtain an upper bound on the Bayesian posterior.
>
> We will add [1] to the prior work section when updates to the manuscript become possible again.
>
> [1] Predictive Uncertainty Estimation via Prior Networks, NeurIPS 2018

---

### Public Comment · ~Pranav_Poduval1 · 2019-10-01
**Dropouts are still Bayesian**

Hron et al. pointed out when trying to prove standard Dropouts as Bayesian Approx. we cannot take mixture of Impulses as the posterior, Gal et al. got around this problem by taking mixture of Gaussian Posterior albeit with very small sigma's  ( of the order 10^(-34) ), Hron et al. called this trick, convolutional approach, so I don't think there is any difference between Dropouts and Bayes By Backprop. I hope you can clear this up in your paper. Good Work otherwise

---

> ### Author Response · Authors · 2019-10-10
> **Thanks for the feedback.**
>
> Thanks for the feedback on our paper's introduction Pranav!
>
> You are right that the Bayes by backprop approach can be extended to use approximate posterior distributions which are non-Gaussian. Although Monte Carlo dropout can be viewed in this light, to do so requires a rather unnatural approximating family from the perspective approximate inference. It then requires a limit to be taken (like the convolutional one you and Hron point out) or Hron's generalisation of variational inference to a quasi-KL. An alternative view of MC dropout is as an ensemble method in which the ensemble members have shared parameters (meaning that they should therefore be trained together) and where the ensembling is applied at test time too. This latter view is arguably as natural as the Bayesian interpretation. For this reason we set MC dropout apart in the discussion of prior work.
>
> We take your valuable feedback on board and will make these points -- including the connection between MC dropout and Bayes by backprop  -- explicit in the updated version of the paper.

---

### Public Comment · ~Khurai_Kim1 · 2019-11-06
**About structural priors implied by the architectures**

I think you could include [1][2] when you say "we are implicitly making the assumption that the network architecture is appropriate for the dataset anyway."

[1] Ulyanov, Dmitry, Andrea Vedaldi, and Victor Lempitsky. "Deep image prior." Proceedings of the IEEE Conference on Computer Vision and Pattern Recognition. 2018.
[2] Zezhou Cheng, Matheus Gadelha, Subhransu Maji, Daniel Sheldon. "A Bayesian Perspective on the Deep Image Prior".  CVPR 2019.

These explicitly discuss the subject of structural priors of network architectures.

---

> ### Author Response · Authors · 2019-11-08
> **Thanks for the pointers.**
>
> Thanks for mentioning these. We will reference them in the revised paper.

---

### Author Response · Authors · 2019-11-08
**Thanks for the reviews! We will provide a revised version of the paper.**

Thanks to all reviewers for providing feedback!

We address the main points below. We also provide a detailed response to each review individually.

1. [Significance] The main contribution of the paper is that we provide a sound theoretical framework for what was previously an ad-hoc method lacking principled justification, but was found to work really well in practice (Burda et al). Osband et al. only justified the method for linear regression, while our theory works for arbitrary deep networks.

2. [Comparison with deep ensembles] We do not aim do replace deep ensembles, but to provide a method which is complementary to them in settings where conservative uncertainties are desired. Having said that, our it also has practical advantages. In particular, in practice, deep ensembles can become overconfident and poorly calibrated. Also, deep ensembles require using more bootstraps (we will describe this more explicitly in the updated paper). We will provide a more detailed comparison with deep ensembles in a revised version of the paper.

We agree with the reviewers that the clarity of the submission can be improved.
We will provide a revised version of the paper.

References (for this post):
Yuri Burda, Harrison Edwards, Amos Storkey, and Oleg Klimov.  Exploration by random network distillation.
Ian Osband, John Aslanides, and Albin Cassirer. Randomized prior functions for deep reinforcement learning. NeurIPS 2019
Ian Osband, Benjamin Van Roy, Daniel J. Russo, and Zheng Wen. Deep exploration via randomized value func-tions. Journal of Machine Learning Research, 2019.

---

### Author Response · Authors · 2019-11-12
**We have uploaded a revised version.**

We have uploaded a revised version of the paper. We made many clarifications throughout the paper, particularly in Section 4. We list the main points below.

In response to review #1:
- filled in the missing technical details, in particular:
  - we added details of the architecture (Appendix A2).
  - we added information about training error (Appendix A2).
  - we added a description of weight cloning (last paragraph, Section 5)
- added clarifications concerning dropout

In response to reviewer #2:
- we extended our discussion of the prior
- we enhanced a discussion of deep ensembles, adding an example where they can fail (each model in an ensemble convex in weights, which means all models converge to the same solution).
- we shortened the preliminaries and used the space for a clearer description of our method

In response to review #3:
- we added an explicit statement (Corollary 1) that shows conservatism for finite number of bootstraps, including just one bootstrap.
- we added an additional appendix that lists the variances of random variables used in the paper (Appendix E).
- we changed the notations to be more readable ('v' symbol for variance).
- Y axis in Figure 3,6 has been changed as requested

We are still working on the paper and will keep updating it.

---

### Author Response · Authors · 2019-11-15
**Final version of paper.**

Thanks again to all reviewers. We have now uploaded the final version of the paper.

Overall, we feel that main contribution of the paper - a sound theoretical framework for what was previously an ad-hoc method lacking principled justification will prove useful for the community.

Relative to the original submission, we added a new experiment on CIFAR100 (appendix B) and an ablation on initialization scale (page 10). We now include details of the classification and OOD accuracy as well as the other technical details requested by reviewer 1. We also made many clarifications in the text, requested by the reviewers. Particularly, we provide additional background on when deep ensembles can fail as well as on the used prior.

We provide individual summary response messages below each review.

---

### Decision · Program_Chairs · 2019-12-19

**Decision:**

Accept (Poster)

**Comment:**

The paper provides theoretical justification for a previously proposed method for uncertainty estimation based on sampling from a prior distribution (Osband et al., Burda et al.).

The reviewers initially raised concerns about significance, clarity and experimental evaluation, but the author rebuttal addressed most of these concerns.

In the end, all the reviewers agreed that the paper deserves to be accepted.